# Reliability assessment model for multiple stress factors accelerated degradation test using a Wiener process with random effects

Qianqian Huang[1,2], Jiayin Tang [3]*, Xuefeng Feng[3]

**1** College of Computer Science and Artificial Intelligence, Southwest Minzu University, Chengdu, Sichuan, China, **2** The Key Laboratory for Computer Systems of State Ethnic Affairs Commission, Southwest Minzu University, Chengdu, Sichuan, China, **3** Department of Statistics, School of Mathematics, Southwest Jiaotong University, Chengdu, Sichuan, China

* jiayintang@163.com

**Data availability statement:** The datasets used and/or analyzed during the current study are

## Abstract

In practical applications, products are usually exposed to multiple stress factors (including environmental stresses and operating stresses) simultaneously. However, existing work on accelerated degradation test mainly focuses on the case of a single stress factor. This motivates the need to develop a reliability assessment model for accelerated degradation test involving multiple stress factors. Therefore, this paper proposes a Wiener process-based accelerated degradation test model that simultaneously considers multiple stress factors, random effects and measurement errors. Then the explicit expression for the lifetime distribution under normal operating conditions of the proposed Wiener accelerated degradation test model is obtained, along with its approximate mean lifetime. In addition, the maximum likelihood estimates of model parameters are derived using the profile likelihood approach, and maximum likelihood estimates for some reliability metrics under normal operating conditions are also obtained. Besides, we construct confidence intervals for model parameters and some reliability metrics using the bias-corrected and accelerated percentile bootstrap method. Finally, the performance of the proposed method is demonstrated by extensive simulation studies, and a numerical example.

## Acronyms and abbreviations

ADT accelerated degradation test
AIC akaike information criterion
AW average width
BCa bias-corrected and accelerated
CSADT constant-stress accelerated degradation test
CP coverage probability
EM expectation maximization

available from the Data Access Committee of Sichuan Province Applied Statistics Society (sichuanshengxctjxh@163.com) upon reasonable request.

**Funding:** This work was supported by the National Social Science Fund of China (Grant No. 23BTJ010), the funded researcher (JiaYin Tang) modeling concept, writing-review & editing, funding acquisition. This work was also supported by a grant from supported by the Southwest Minzu University Research Startup Funds (Grant No. RQD2023017) and by the Fundamental Research Funds for the Central Universities in Southwest Minzu University (ZYN2025067), the funded researcher (Qianqian Huang) Writing original draft, Methodology, Conceptualization, Investigation, Formal Analysis, Funding acquisition.

**Competing interests:** The authors have declared that no competing interests exist.

| | |
|---|---|
| Log-LF | log-likelihood function |
| MCSADT | multiple constant-stress accelerated degradation test |
| MLEs | maximum likelihood estimates |
| SSADT | step-stress accelerated degradation test |
| PSADT | progressive-stress accelerated degradation test |
| RMSE | Relative mean square error |

## Introduction

The accelerated degradation test (ADT) methods are becoming increasingly popular for quickly evaluating the reliability of highly reliable and long life products, especially during the product development stages. In an ADT, products are subjected to harsher-than-normal operating conditions to expedite the failure processes, and their measurable performance or quality characteristics are observed at regular time intervals. Then, the degradation information at higher levels of one or more accelerating variables (e.g., temperature, humidity, voltage or pressure) is extrapolated, through a physically appropriate statistical model, to obtain estimates of reliability metrics of products under normal operating conditions (see, for example, Limon et al. [1], Lee and Tang [2]). ADTs can typically be classified into constant-stress ADT (CSADT), step-stress ADT (SSADT), and progressive-stress ADT (PSADT) based on different stress loading methods. More detailed descriptions of ADTs can be found in Nelson [3], and Meeker and Escobar [4].

ADTs are commonly used to obtain more lifetime information in a relatively short testing duration and can greatly reduce testing cost, thus research on ADTs has attracted much attention in recent years. In literature, degradation models for ADT data can mainly be classified into general path models and stochastic process models. In a general path model, a regression model is adopted to depict the common degradation properties among units, and the parameters in the model are usually randomized to characterize the unit-to-unit variability. Lu and Meeker [5] first proposed a non-linear mixed-effects model to characterize degradation trajectories. Some developments of general path models have been made by researchers such as Lu et al. [6], Meeker et al. [7], Bae et al. [8], Yuan and Pandey [9], and Shi and Meeker [10]. Although general path models can characterize the unit-to-unit variability, they cannot model the temporal variability of the degradation over time, as pointed out by Pandey et al. [11]. To address this issue, stochastic process models are introduced to the area of degradation-based reliability modelling and analysis.

In degradation data analysis, three popular stochastic process models are the Wiener process, the Gamma process, and the inverse Gaussian process, see for example, Si et al. [12], Li et al. [13], Wang et al. [14], Duan and Wang [15], Zheng et al. [16], and Tseng et al. [17]. Among these models, the Wiener process model has been extensively used in recent years to analyse accelerated degradation data due to its attractive mathematical properties and physical interpretations. In recent years, for example, Guan et al. [18] have carried out the statistical inference of the model parameters for the Wiener process CSADT model using an objective Bayesian method. Jiang et al. [19] investigated the inferential methods for interval estimation of a Wiener process CSADT model and proposed the exact confidence intervals for model parameters, generalized confidence intervals and prediction intervals for some reliability metrics under normal operating conditions. He and Tao [20] proposed a dual CSADT model based on Wiener processes, using an objective Bayesian approach to make statistical inference on unknown parameters of the proposed model. Sun et al. [21] developed a multivariate dependent model for accelerated degradation tests based on a general Wiener

process with random effect, and a D-vine copula function was used to represent the correlations between the degradations. Liu et al. [22] proposed a multi-objective optimization method for the accelerated degradation test based on the Wiener process. He [23] studied an objective Bayesian method to analyse the accelerated degradation model based on the new Wiener process proposed by Ye et al. [24].

In practical applications, products are often exposed to several stress factors simultaneously, including temperature, humidity, vibration, voltage, and more. However, existing works on ADT modelling mainly focus on the case of a single stress factor. This may not be practical when the quality characteristics of the product have slow degradation rates. To overcome this difficulty, some researchers have made contributions to the reliability modelling and assessment of ADT with multiple stress factors. Tsai et al. [25] proposed an algorithm to determine an optimal strategy for the Wiener process CSADT with two accelerated stress factors. Tsai et al. [26] investigated the statistical inference and optimal design of a gamma CSADT model with two accelerating variables, and proposed an algorithm to achieve an optimal plan of the gamma CSADT model. Tung and Tseng [27] presented an analytical approach of applying general equivalence theorem to determine the optimal stress-level combinations together with their optimal sample-size allocations of a gamma process CSADT with two accelerating variables. Li et al. [28] investigated a Wiener process CSADT model with random effects that takes into account several acceleration variables, and proposed an expectation maximization (EM) algorithm for estimating the unknown parameters. Limon et al. [29] proposed an optimal design approach for a CSADT model based on the gamma degradation process with multi-stress factors and interaction effects.

Consequently, this paper investigates a reliability assessment model for a Wiener multiple constant-stress ADT (MCSADT) that simultaneously considers temporal variability, unit-to-unit variability, and measurement variability. This degradation model incorporates multiple simultaneous stress factors (e.g., environmental and operational stresses), thereby better reflecting real-world conditions and enhancing the realism and robustness of the degradation analysis through the inclusion of random effects and measurement errors. Besides, maximum likelihood estimates of the model parameters are derived using the profile likelihood approach, which is effective in handling complex models and provides reliable estimates. Finally, the confidence intervals for the parameters of interest are obtained using the bias-corrected and accelerated (BCa) percentile bootstrap method, which provides enhanced inference accuracy and demonstrates particular robustness in small-sample scenarios.

The remainder of this paper is organized as follows. Section 2 presents the Wiener process MCSADT model with random effects and measurement errors,as well as the associated assumptions. Section 3 provides the maximum likelihood estimates of unknown parameters and some reliability metrics of the proposed MCSADT model. In section 4, bootstrap confidence intervals are provided for unknown parameters and some reliability metrics under normal stress level combination. In section 5, simulation studies and numerical example were carried out to demonstrate the performance of the proposed Wiener MCSADT model. Finally, some concluding remarks are presented in Section 6.

# 1 Wiener process MCSADT model

## 1.1 Wiener process with random effects

Motivated by real examples, a Wiener process-based degradation model with random effect can be represented as

$$X(t) = X(0) + \beta t + \sigma_B B(t), \ \beta \sim N(\mu_\beta, \sigma_\beta^2), \tag{1}$$

where $X(t)$ denotes the true degradation of the quality characteristic of products at time $t$ ($t \geq 0$), and $X(0) = x_0$ is a known initial degradation. Without loss of generality, we suppose that $X(0) = x_0 = 0$ in the following. The drift parameter $\beta$ with $N(\mu_\beta, \sigma_\beta^2)$ represents the unit-to-unit variability, while the diffusion parameter $\sigma_B > 0$ describes the common degradation feature for all products in the same batch. $B(t)$ is the standard Brownian motion, which represents the temporal variability.

It is noteworthy that we have $P(\beta \leq 0) \approx 0$ when $\mu_\beta \gg \sigma_\beta$ based on the normal assumption $\beta \sim N(\mu_\beta, \sigma_\beta^2)$, we thus assume $\mu_\beta \gg \sigma_\beta$. The ideas of incorporating random effects and normal distribution assumptions are widely adopted in degradation modelling literature, see [12,30–33] for example.

In realistic applications, obtaining accurate performance degradation data for products is quite challenging. Instead, observed quality characteristics are inevitably affected by measurement errors stemming from disturbances, noise, non-ideal instruments, and other factors. Therefore, to account for the effect of measurement variability in degradation modelling, the observed process $\{Y(t), t \geq 0\}$, which describes the degradation paths of products over time, can be formulated as

$$M_0: \ Y(t) = X(t) + \varepsilon = \beta t + \sigma_B B(t) + \varepsilon, \ \beta \sim N(\mu_\beta, \sigma_\beta^2), \tag{2}$$

where $\varepsilon$ represents random measurement errors, assumed to be statistically independent and identically distributed (*i.i.d.*) with $\varepsilon \sim N(0, \sigma_\varepsilon^2)$ at any time point $t$. It is further assumed that $\beta$, $B(t)$ and $\varepsilon$ are mutually statistically independent. Our model offers the advantage of simultaneously considering temporal variability, unit-to-unit variability, and measurement variability.

The degradation model $M_0$ encompasses the following widely used models that have been extensively studied in the literature as limiting cases.

(1) Assuming that $\sigma_\varepsilon = 0$, model $M_0$ reduces to the linear random effects Wiener process degradation model without measurement errors, as described in [32–34].
(2) Letting $\sigma_\beta = 0$ and $\sigma_\varepsilon = 0$, model $M_0$ turns to the linear fixed effect Wiener process degradation model with measurement errors in [18,19,35,36], where only the temporal variability is considered.

### 1.2 MCSADT for Wiener process with random effect

In this paper, we consider a MCSADT model based on the Wiener process with random effect, incorporating $p$ accelerating variables and $l$ stress level combinations. The description of the MCSADT is presented as follows.

Suppose devices are tested at $l$ stress level combinations (i.e., test group), wherein, for $i = 1, 2, ..., l$, there are $n_i$ identical specimens are subjected to each test group $\boldsymbol{x}_i = (x_{i1}, x_{i2}, ..., x_{ip})'$, where $x_{iq}$ denotes the $i$-th (transformed) stress level of the $q$-th stress factor (i.e., accelerating variable). In $i$-th test group, each device undergoes a degradation test at inspection times $t_{ij1}, t_{ij2}, ..., t_{ijm_{ij}}$, respectively. The corresponding performance characteristics for each unit are collected, where $t_{ijk}$ represents the $k$-th inspection time of the $j$-th test unit in $i$-th test group, with $j = 1, 2, ..., n_i$ and $k = 1, 2, ..., m_{ij}$.

The statistical inference and reliability assessment for the Wiener process MCSADT model with random effect, the following three basic assumptions are necessary.

(A1) Under test group $x_i$, the degradation paths of units follow a Wiener process with the drift parameter $\beta_i$, and diffusion parameter $\sigma_{i,B}$, i.e.,

$$X_i(t) = X(t|x_i) = \beta_i t + \sigma_{i,B} B(t), \ \beta_i \sim N(\mu_{i,\beta}, \sigma_{i,\beta}^2), \ i = 0, 1, 2, ..., l. \tag{3}$$

(A2) $\sigma_{0,B} = \sigma_{1,B} = \cdots = \sigma_{l,B} = \sigma_B$. This means that the degradation mechanism are statistically the same under different test group $x_i$, $i = 1, 2, ..., l$.

(A3) The drift parameter $\beta_i$ can be expressed as the following random log-linear relationship

$$\beta_i = a \exp(\gamma_1 x_{i1} + \gamma_2 x_{i2} + \cdots + \gamma_p x_{ip}), \ a \sim N(\mu_a, \sigma_a^2), \ i = 0, 1, ..., l, \tag{4}$$

where $\gamma_1, \gamma_2, ..., \gamma_p$ are unknown parameters related to the device and test method, $x_{i1}, x_{i2}, ..., x_{ip}$ are transformed variables.

From the assumption A3, we can conclude that $\beta_i \sim N(\mu_{i,\beta}, \sigma_{i,\beta}^2)$, where $\mu_{i,\beta} = \mu_a \lambda(x_i)$, $\sigma_{i,\beta}^2 = \sigma_a^2 \lambda^2(x_i)$, and $\lambda(x_i) = \exp(\gamma_1 x_{i1} + \gamma_2 x_{i2} + \cdots + \gamma_p x_{ip})$.

## 1.3 Failure time distribution

In this subsection, we present the failure time distribution (FTD) of the Wiener process with random effects at normal stress level combination. The FTD is critically significant for conducting statistical inference and assessing reliability.

Let $\omega$ be the failure threshold of product at normal stress level combination, and the degradation process is assumed to be monotonic increasing over time. We adopt the first hitting time (FHT) concept of stochastic process to define the lifetime. Therefore, the lifetime $T$ of a product at normal stress level combination $x_0$ is defined as the FHT of the Wiener process $X_0(t)$ reaches the failure threshold $\omega$, i.e., the lifetime $T$ can be expressed as

$$T = \inf\left\{t : X_0(t) \geq \omega \big| X_0(0) < \omega\right\}. \tag{5}$$

It is well known that the FHT distribution of the Wiener process with random effect follows an inverse Gaussian distribution [30]. We thus obtain the probability density function (PDF) and cumulative distribution function (CDF) of the lifetime $T$ as follows

$$f_T(t) = \frac{\omega}{\sqrt{2\pi t^3(\sigma_{0,\beta}^2 t + \sigma_B^2)}} \exp\left\{-\frac{(\omega - \mu_{0,\beta} t)^2}{2t(\sigma_{0,\beta}^2 t + \sigma_B^2)}\right\}, \tag{6}$$

$$F_T(t) = \Phi\left(\frac{\mu_{0,\beta} t - \omega}{\sqrt{\sigma_B^2 t + \sigma_{0,\beta}^2 t^2}}\right) + \exp\left(\frac{2\omega\mu_{0,\beta}}{\sigma_B^2} + \frac{2\omega^2 \sigma_{0,\beta}^2}{\sigma_B^4}\right) \Phi\left(-\frac{\mu_{0,\beta}\sigma_B^2 t + (2\sigma_{0,\beta}^2 t + \sigma_B^2)\omega}{\sigma_B^2 \sqrt{\sigma_B^2 t + \sigma_{0,\beta}^2 t^2}}\right), \tag{7}$$

where $\Phi(\cdot)$ denotes the CDF for the standard normal distribution.

According to the result of Peng and Tseng [30], the mean of the lifetime $T$ at normal stress level combination $x_0$, referred to $t_{0,MTTF}$, can be calculated as follows

$$t_{0,MTTF} = E(T) = E\left(\frac{\omega}{\beta_0}\right) = \frac{\omega}{\sigma_{0,\beta}^2} \exp\left(-\frac{\mu_{0,\beta}^2}{\sigma_{0,\beta}^2}\right) \int_0^{\mu_{0,\beta}} \exp\left(\frac{x^2}{\sigma_{0,\beta}^2}\right) dx = \frac{\sqrt{2}\omega}{\sigma_{0,\beta}} D\left(\frac{\mu_{0,\beta}}{\sqrt{2}\sigma_{0,\beta}}\right), \tag{8}$$

where $D(x) = \exp(-x^2)\int_0^x \exp(-u^2)\mathrm{d}u$ is the Dawson integral. Besides, according the property of Dawson's integral, $D(x) \approx \frac{1}{2x}$ for large $x$, under the assumption $\mu_{0,\beta} \gg \sigma_{0,\beta}$, we have $t_{0,MTTF} = E(T) \approx \frac{\omega}{\mu_{0,\beta}}$.

## 2 Point estimation for Wiener MCSADT model parameters

In this section, we present the maximum likelihood estimates (MLEs) of the unknown parameters for the proposed Wiener MCSADT model. Using the invariance property of MLE, we then derive the MLEs of some reliability metrics at normal stress level combination.

Assume that $n_i$ units are tested at stress level combination $\boldsymbol{x}_i$, and the degradation measurements for the $j$-th test unit are available at inspection times $t_{ij1}, t_{ij2}, ..., t_{ijm_{ij}}$, where $m_{ij}$ is the number of measurements of the $j$-th test unit, $i = 1, 2, ..., l, j = 1, 2, ..., n_i$. The observed performance characteristic of the $j$th test unit at time $t_{ijk}$ at stress level combination $\boldsymbol{x}_i$ can be expressed as

$$Y_{ij}(t_{ijk}) = \beta_{ij}t_{ijk} + \sigma_B B(t_{ijk}) + \varepsilon_{ijk}, \ i = 1, 2, ..., l, \ j = 1, 2, ..., n_i, \ k = 1, 2, ..., m_{ij},$$

where $\beta_{ij} \sim N(\mu_{i,\beta}, \sigma_{i,\beta}^2)$, $\varepsilon_{ijk} \overset{i.i.d}{\sim} N(0, \sigma_\varepsilon^2)$. In addition, the $\beta_{ij}$, $B(t_{ijk})$, and $\varepsilon_{ijk}$ are assumed to be mutually statistically independent.

For simplicity, let $y_{ijk} = Y_{ij}(t_{ijk})$, $\boldsymbol{t}_{ij} = (t_{ij1}, t_{ij2}, ..., t_{ijm_{ij}})'$, $\boldsymbol{y}_{ij} = (y_{ij1}, y_{ij2}, ..., y_{ijm_{ij}})'$, and $\boldsymbol{y} = (\boldsymbol{y}_{11}', \boldsymbol{y}_{12}', ..., \boldsymbol{y}_{1n_1}', ..., \boldsymbol{y}_{l1}', \boldsymbol{y}_{l2}', ..., \boldsymbol{y}_{ln_l}')'$. Using the property of stationary and independent increments of the Wiener process, $\boldsymbol{y}_{ij}$ follows a multivariate normal with mean $\mu_a \lambda(\boldsymbol{x}_i)\boldsymbol{t}_{ij}$ and variance (see, for instance [30])

$$\boldsymbol{\Sigma}_{ij} = \sigma_a^2 \lambda^2(\boldsymbol{x}_i)\boldsymbol{t}_{ij}\boldsymbol{t}_{ij}' + \boldsymbol{\Omega}_{ij}, \tag{9}$$

that is, $\boldsymbol{y}_{ij} \sim N_{m_{ij}}\left(\mu_a \lambda(\boldsymbol{x}_i)\boldsymbol{t}_{ij}, \boldsymbol{\Sigma}_{ij}\right)$, where

$$
\begin{aligned}
\lambda(\boldsymbol{x}_i) &= \exp(\gamma_1 x_{i1} + \gamma_2 x_{i2} + \cdots + \gamma_p x_{ip}) \\
\boldsymbol{\Omega}_{ij} &= \sigma_B^2 \boldsymbol{Q}_{ij} + \sigma_\varepsilon^2 \boldsymbol{I}_{m_{ij}}, \\
\boldsymbol{Q}_{ij} &= \begin{pmatrix}
t_{ij1} & t_{ij1} & \cdots & t_{ij1} \\
t_{ij1} & t_{ij2} & \cdots & t_{ij2} \\
\vdots & \vdots & \ddots & \vdots \\
t_{ij1} & t_{ij2} & \cdots & t_{ijm_{ij}}
\end{pmatrix},
\end{aligned}
\tag{10}
$$

and $\boldsymbol{I}_{m_{ij}}$ is an identity matrix of order $m_{ij}$, $i = 1, 2, ..., l, j = 1, 2, ..., n_i$.

Based on the three basic assumptions in section 2, and the performance degradation data $\boldsymbol{y}$, the log-likelihood function for $\theta$ can be expressed by

$$L(\boldsymbol{\theta}|\boldsymbol{y}) = \prod_{i=1}^{l}\prod_{j=1}^{n_i} \frac{1}{(2\pi)^{m_{ij}/2}|\boldsymbol{\Sigma}_{ij}|^{1/2}} \exp\left\{-\frac{1}{2}(\boldsymbol{y}_{ij} - \mu_a\lambda(\boldsymbol{x}_i)\boldsymbol{t}_{ij})'\boldsymbol{\Sigma}_{ij}^{-1}(\boldsymbol{y}_{ij} - \mu_a\lambda(\boldsymbol{x}_i)\boldsymbol{t}_{ij})\right\}. \tag{11}$$

From (11), the associated log-likelihood function is given by

$$\ell(\boldsymbol{\theta}|\boldsymbol{y}) = -\frac{N\ln(2\pi)}{2} - \frac{1}{2}\sum_{i=1}^{l}\sum_{j=1}^{n_i}\ln|\boldsymbol{\Sigma}_{ij}| - \frac{1}{2}\sum_{i=1}^{l}\sum_{j=1}^{n_i}(\boldsymbol{y}_{ij} - \mu_a\lambda(\boldsymbol{x}_i)\boldsymbol{t}_{ij})'\boldsymbol{\Sigma}_{ij}^{-1}(\boldsymbol{y}_{ij} - \mu_a\lambda(\boldsymbol{x}_i)\boldsymbol{t}_{ij}), \tag{12}$$

where $\boldsymbol{\theta} = (\mu_a, \sigma_a^2, \sigma_B^2, \sigma_\varepsilon^2, \gamma_1, \gamma_2, ..., \gamma_p)$ is the unknown parameter vector, $N = \sum_{i=1}^l \sum_{j=1}^{n_i} m_{ij}$.

To estimate the unknown parameter $\boldsymbol{\theta}$, we first re-parameterize by $\widetilde{\sigma}_B^2 = \sigma_B^2/\sigma_a^2$, $\widetilde{\sigma}_\varepsilon^2 = \sigma_\varepsilon^2/\sigma_a^2$, and $\widetilde{\boldsymbol{\Sigma}}_{ij} = \boldsymbol{\Sigma}_{ij}/\sigma_a^2$. Therefore, one can obtain the log-likelihood function of $\boldsymbol{\theta}$ as

$$\ell(\tilde{\boldsymbol{\theta}}|\boldsymbol{y}) = -\frac{N}{2}\left(\ln(2\pi) + \ln\sigma_a^2\right) - \frac{1}{2}\sum_{i=1}^l\sum_{j=1}^{n_i}\ln|\widetilde{\boldsymbol{\Sigma}}_{ij}| -$$

$$\frac{1}{2\sigma_a^2}\sum_{i=1}^l\sum_{j=1}^{n_i}\left(\boldsymbol{y}_{ij} - \mu_a\lambda(\boldsymbol{x}_i)\boldsymbol{t}_{ij}\right)'\widetilde{\boldsymbol{\Sigma}}_{ij}^{-1}\left(\boldsymbol{y}_{ij} - \mu_a\lambda(\boldsymbol{x}_i)\boldsymbol{t}_{ij}\right) \quad (13)$$

where $\tilde{\boldsymbol{\theta}} = (\mu_a, \sigma_a^2, \tilde{\sigma}_B^2, \tilde{\sigma}_\varepsilon^2, \gamma_1, \gamma_2, ..., \gamma_p)$. To simplify the log-likelihood function (13), we adopt the following two well-known results

$$|\widetilde{\boldsymbol{\Sigma}}_{ij}| = |\widetilde{\boldsymbol{\Omega}}_{ij}|\left(1 + \lambda^2(\boldsymbol{x}_i)\boldsymbol{t}_{ij}'\widetilde{\boldsymbol{\Omega}}_{ij}^{-1}\boldsymbol{t}_{ij}\right), \; i = 1, 2, ..., l, \; j = 1, 2, ..., n_i,$$

and

$$\widetilde{\boldsymbol{\Sigma}}_{ij}^{-1} = \widetilde{\boldsymbol{\Omega}}_{ij}^{-1} - \frac{\lambda^2(\boldsymbol{x}_i)}{1 + \lambda^2(\boldsymbol{x}_i)\boldsymbol{t}_{ij}'\widetilde{\boldsymbol{\Omega}}_{ij}^{-1}\boldsymbol{t}_{ij}}\widetilde{\boldsymbol{\Omega}}_{ij}^{-1}\boldsymbol{t}_{ij}\boldsymbol{t}_{ij}'\widetilde{\boldsymbol{\Omega}}_{ij}^{-1}, \; i = 1, 2, ..., l, \; j = 1, 2, ..., n_i.$$

where $\widetilde{\boldsymbol{\Omega}}_{ij} = \widetilde{\sigma}_B^2\boldsymbol{Q}_{ij} + \widetilde{\sigma}_\varepsilon^2\boldsymbol{I}_{m_{ij}}$.

Taking the first order derivatives of the $\ell(\tilde{\boldsymbol{\theta}}|\boldsymbol{y})$ with respect to $\mu_a, \sigma_a^2$, one have

$$\frac{\partial\ell(\tilde{\boldsymbol{\theta}}|\boldsymbol{y})}{\partial\mu_a} = \frac{1}{\sigma_a^2}\left(\sum_{i=1}^l\sum_{j=1}^{n_i}\lambda(\boldsymbol{x}_i)\boldsymbol{t}_{ij}'\widetilde{\boldsymbol{\Sigma}}_{ij}^{-1}\boldsymbol{y}_{ij} - \mu_a\sum_{i=1}^l\sum_{j=1}^{n_i}\lambda^2(\boldsymbol{x}_i)\boldsymbol{t}_{ij}'\widetilde{\boldsymbol{\Sigma}}_{ij}^{-1}\boldsymbol{t}_{ij}\right),$$

$$\frac{\partial\ell(\tilde{\boldsymbol{\theta}}|\boldsymbol{y})}{\partial\sigma_a^2} = -\frac{N}{2\sigma_a^2} + \frac{1}{2(\sigma_a^2)^2}\sum_{i=1}^l\sum_{j=1}^{n_i}\left(\boldsymbol{y}_{ij} - \mu_a\lambda(\boldsymbol{x}_i)\boldsymbol{t}_{ij}\right)'\widetilde{\boldsymbol{\Sigma}}_{ij}^{-1}\left(\boldsymbol{y}_{ij} - \mu_a\lambda(\boldsymbol{x}_i)\boldsymbol{t}_{ij}\right).$$

Therefore, for specified values of $\acute{\boldsymbol{\theta}} = (\tilde{\sigma}_B^2, \tilde{\sigma}_\varepsilon^2, \gamma_1, \gamma_2, ..., \gamma_p)$, the restricted MLEs (RMLEs) of $\mu_a$ and $\sigma_a^2$, can be presented as

$$\hat{\mu}_a(\acute{\boldsymbol{\theta}}) = \frac{\sum_{i=1}^l\sum_{j=1}^{n_i}\lambda(\boldsymbol{x}_i)\boldsymbol{t}_{ij}'\widetilde{\boldsymbol{\Sigma}}_{ij}^{-1}\boldsymbol{y}_{ij}}{\sum_{i=1}^l\sum_{j=1}^{n_i}\lambda^2(\boldsymbol{x}_i)\boldsymbol{t}_{ij}'\widetilde{\boldsymbol{\Sigma}}_{ij}^{-1}\boldsymbol{t}_{ij}}, \quad (14)$$

$$\hat{\sigma}_a^2(\acute{\boldsymbol{\theta}}) = \frac{1}{N}\sum_{i=1}^l\sum_{j=1}^{n_i}\left(\boldsymbol{y}_{ij} - \hat{\mu}_a(\acute{\boldsymbol{\theta}})\lambda(\boldsymbol{x}_i)\boldsymbol{t}_{ij}\right)'\widetilde{\boldsymbol{\Sigma}}_{ij}^{-1}\left(\boldsymbol{y}_{ij} - \hat{\mu}_a(\acute{\boldsymbol{\theta}})\lambda(\boldsymbol{x}_i)\boldsymbol{t}_{ij}\right). \quad (15)$$

Substituting (14) and (15) into (13), one can then get the profile log-likelihood function as

$$\ell(\acute{\boldsymbol{\theta}}|\boldsymbol{y}) = -\frac{N}{2}(\ln(2\pi) + 1) - \frac{N}{2}\ln\left(\hat{\sigma}_a^2(\acute{\boldsymbol{\theta}})\right) - \frac{1}{2}\sum_{i=1}^l\sum_{j=1}^{n_i}\ln|\widetilde{\boldsymbol{\Sigma}}_{ij}|. \quad (16)$$

The MLEs of $\acute{\boldsymbol{\theta}}$, denoted as $\hat{\acute{\boldsymbol{\theta}}} = (\hat{\tilde{\sigma}}_B^2, \hat{\tilde{\sigma}}_\varepsilon^2, \hat{\gamma}_1, \hat{\gamma}_2, ..., \hat{\gamma}_p)$, can be derived by maximizing the profile log-likelihood function (16) through a multi-dimensional optimization algorithm, such as the Nelder-Mead algorithm. Subsequently, substitute $\hat{\acute{\boldsymbol{\theta}}}$ into (14) and (15), one can obtain the

MLEs for $\mu_a, \sigma_a^2$, denoted as $\hat{\mu}_a = \hat{\mu}_a(\hat{\hat{\theta}}), \hat{\sigma}_a^2 = \hat{\sigma}_a^2(\hat{\hat{\theta}})$, respectively. In addition, the MLEs for $\sigma_B^2, \sigma_\varepsilon^2$ are given by $\hat{\sigma}_B^2 = \hat{\hat{\sigma}}_B^2 \hat{\sigma}_a^2, \hat{\sigma}_\varepsilon^2 = \hat{\hat{\sigma}}_\varepsilon^2 \hat{\sigma}_a^2$, respectively.

Thereafter, using the invariance property of MLE, the MLEs of mean lifetime, reliability function at mission time $t_0$, and the $p$th ($0<p<1$) quantile of a product under normal operating conditions $\boldsymbol{x}_0$, could be estimated as follows

$$\hat{t}_{0,MTTF} \approx \frac{\omega}{\hat{\mu}_{0,\beta}},$$

$$\hat{R}(t_0) = 1 - \hat{F}_T(t|\hat{\mu}_{0,\beta}, \hat{\sigma}_{0,\beta}, \hat{\sigma}_B),$$

$$\hat{t}_{0,p} = \hat{F}_T^{-1}(p|\hat{\mu}_{0,\beta}, \hat{\sigma}_{0,\beta}, \hat{\sigma}_B),$$

where $\hat{\mu}_{0,\beta} = \hat{\mu}_a \hat{\lambda}(\boldsymbol{x}_0), \hat{\sigma}_{0,\beta} = \hat{\sigma}_a \hat{\lambda}(\boldsymbol{x}_0), \hat{\lambda}(\boldsymbol{x}_0) = \exp(\hat{\gamma}_1 x_{i1} + \hat{\gamma}_2 x_{i2} + \cdots + \hat{\gamma}_p x_{ip})$.

## 3 Confidence intervals

Since the MLEs of model parameters are not explicit expressions, it is impossible to obtain their distributions, as well as the corresponding exact confidence intervals. We will, therefore, discuss bootstrap confidence intervals for the model parameters in this section, employing the bias-corrected and accelerated (BCa) percentile method (see Efron and Tibshirani [37] (pp. 184-188)).

To obtain the BCa percentile bootstrap confidence intervals for $\mu_a, \sigma_a^2, \sigma_B^2, \sigma_\varepsilon^2, \gamma_1, \gamma_2, ..., \gamma_p$, and the parameters of interest $t_{0,MTTF}, R(t_0), t_{0,p}$, we adopt the following two algorithms.

(1) Based on the original sample $\boldsymbol{y} = (\boldsymbol{y}_{11}', \boldsymbol{y}_{12}', ..., \boldsymbol{y}_{1n_1}', ..., \boldsymbol{y}_{l1}', \boldsymbol{y}_{l2}', ..., \boldsymbol{y}_{ln_l}')'$,obtain the MLEs $\hat{\mu}_a, \hat{\sigma}_a^2, \hat{\sigma}_B^2, \hat{\sigma}_\varepsilon^2, \hat{\gamma}_1, \hat{\gamma}_2, ..., \hat{\gamma}_p$ of the corresponding parameters. In addition, obtain the MLEs $\hat{t}_{0,MTTF}, \hat{R}_0(t_0), \hat{t}_{0,p}$ for the corresponding parameter of interest using the invariance property of MLE, denoted as $\widehat{\Theta} = (\hat{\mu}_a, \hat{\sigma}_a^2, \hat{\sigma}_B^2, \hat{\sigma}_\varepsilon^2, \hat{\gamma}_1, \hat{\gamma}_2, ..., \hat{\gamma}_p, \hat{t}_{0,MTTF}, \hat{R}_0(t_0), \hat{t}_{0,p})' = (\hat{\theta}_1, \hat{\theta}_2, \hat{\theta}_3, \hat{\theta}_4, \hat{\theta}_5, \hat{\theta}_6, ..., \hat{\theta}_{p+4}, \hat{\theta}_{p+5}, \hat{\theta}_{p+6}, \hat{\theta}_{p+7})'$.

(2) Generate accelerated performance degradation data from a sample of size $n$ from a Wiener MCSADT model using $\hat{\mu}_a, \hat{\sigma}_a^2, \hat{\sigma}_B^2, \hat{\sigma}_\varepsilon^2, \hat{\gamma}_1, \hat{\gamma}_2, ..., \hat{\gamma}_p$ as the true parameters.

(3) Estimate the parameter vector $\boldsymbol{\Theta} = (\mu_a, \sigma_a^2, \sigma_B^2, \sigma_\varepsilon^2, \gamma_1, \gamma_2, ..., \gamma_p, t_{0,MTTF}, R_0(t_0), t_{0,p})' = (\theta_1, \theta_2, \theta_3, \theta_4, \theta_5, \theta_6, ..., \theta_{p+4}, \theta_{p+5}, \theta_{p+6}, \theta_{p+7})'$ based on the new generated performance degradation data set.

(4) Repeat steps 2 and 3, $B$ times, and obtain bootstrap MLEs $\widehat{\Theta}^{(1)}, \widehat{\Theta}^{(2)}, ..., \widehat{\Theta}^{(B)}$, where $\widehat{\Theta}^{(b)} = (\hat{\mu}_a^{(b)}, \hat{\sigma}_a^{2(b)}, \hat{\sigma}_B^{2(b)}, \hat{\sigma}_\varepsilon^{2(b)}, \hat{\gamma}_1^{(b)}, \hat{\gamma}_2^{(b)}, ..., \hat{\gamma}_p^{(b)}, \hat{t}_{0,MTTF}^{(b)}, \hat{R}_0^{(b)}(t_0), \hat{t}_{0,p}^{(b)})' = (\hat{\theta}_1^{(b)}, \hat{\theta}_2^{(b)}, \hat{\theta}_3^{(b)}, \hat{\theta}_4^{(b)}, \hat{\theta}_5^{(b)}, \hat{\theta}_6^{(b)}, ..., \hat{\theta}_{p+4}^{(b)}, \hat{\theta}_{p+5}^{(b)}, \hat{\theta}_{p+6}^{(b)}, \hat{\theta}_{p+7}^{(b)})', b = 1, 2, ..., B$.

(5) For each parameter $\theta_i$ in the parameter vector $\boldsymbol{\Theta}$, arrange the corresponding bootstrap replicates $\hat{\theta}_i^{(b)}$ in ascending order and obtain $\hat{\theta}_i^{[1]}, \hat{\theta}_i^{[2]}, ..., \hat{\theta}_i^{[B]}$ ($i = 1, 2, ..., p + 7$).

A two-side $100(1 - \alpha)\%$ BCa percentile bootstrap confidence intervals of $\theta_i$ is then expressed by

$$\theta_{iL}^* = \hat{\theta}_i^{([B\alpha_{1i}])}, \theta_{iU}^* = \hat{\theta}_i^{([B(1-\alpha_{2i})])}, \qquad i = 1, 2, ..., p + 7, \tag{17}$$

where

$$\alpha_{1i} = \Phi\left(\hat{z}_{0i} + \frac{\hat{z}_{0i} + z_{\alpha/2}}{1 - \hat{a}_i(z_{0i} + z_{\alpha/2})}\right),$$

and

$$\alpha_{2i} = \Phi\left(\hat{z}_{0i} + \frac{\hat{z}_{0i} + z_{1-\alpha/2}}{1 - \hat{a}_i(\hat{z}_{0i} + z_{1-\alpha/2})}\right).$$

Here, $\Phi(\cdot)$ is the standard normal cumulative distribution function. The value of the bias correction $z_{0i}$ can be estimated as

$$\hat{z}_{0i} = \Phi^{-1}\left(\frac{1}{B}\sum_{b=1}^{B} I\left(\hat{\theta}_i^{(b)} < \hat{\theta}_i\right)\right), \qquad i = 1, 2, ..., p + 7,$$

where $I(\cdot)$ is the indicator function; $\Phi^{-1}(\cdot)$ is the inverse function of the standard normal CDF. The acceleration $a_i$ is estimated by

$$\hat{a}_i = \frac{\sum_{j=1}^{m}\left(\hat{\theta}_{i(\cdot)} - \hat{\theta}_{i(j)}\right)^3}{6\left[\sum_{j=1}^{m}\left(\hat{\theta}_{i(\cdot)} - \hat{\theta}_{i(j)}\right)^2\right]^{3/2}}, \qquad i = 1, 2, ..., p + 7,$$

where $\hat{\theta}_{i(j)}$ denotes the MLE of $\theta_i$, which is based on the original sample with the $j$th performance degradation data of $n$ test units deleted from all test groups (i.e., the jackknife estimate), and

$$\hat{\theta}_{i(\cdot)} = \frac{1}{m}\sum_{j=1}^{m}\hat{\theta}_{i(j)},$$

Here, $m$ denotes the measurement times for each test specimen under all test groups.

## 4 Numerical analysis

### 4.1 Simulation studies

In this section, the extensive Monte Carlo simulation studies are conducted to investigate the performance of the proposed model. Comparative analysis will be performed to highlight the superiority of the proposed model. The proposed model is denoted as $M_0$ in this paper, and the corresponding reference model is described as follows

$$M_1: \ X(t) = \beta t + \sigma_B B(t), \ \beta \sim N(\mu_\beta, \sigma_\beta^2), \tag{18}$$

where $\beta$ represents the random effect among the units, model $M_1$ does not take into account the measurement errors in the degradation model.

For the purpose of illustration, we take the number of acceleration variables as four, i.e., $p = 4$.

The performance of the MLEs for model parameters, and the concerning parameters under normal operating conditions are investigated by the following criteria of quantities:

(1) Relative-bias for point estimate $\hat{\vartheta}$ of parameters $\vartheta = \mu_a, \sigma_a^2, \sigma_B^2, \sigma_\varepsilon^2, \gamma_1, \gamma_2, \gamma_3, \gamma_4, t_{0,MTTF},$
$R_0(t_0), t_{0,p}$, which is computed as $\frac{1}{N}\sum_{i=1}^{N}(\frac{\hat{\vartheta}_i - \vartheta}{\vartheta})$.

(2) Relative mean square error (RMSE) for point estimator $\hat{\vartheta}$ of $\vartheta$, which is calculated by
$\frac{1}{N}\sum_{i=1}^{N}\left(\frac{\hat{\vartheta}_i - \vartheta}{\vartheta}\right)^2$.

(3) Coverage probability (CP) of $100(1 - \alpha)\%$ confidence interval of $\vartheta$, which is defined as the proportion of the times that the estimated interval contains its true value.

(4) Average width (AW) of the $100(1 - \alpha)\%$ CI of $\vartheta$ is defined as the average interval width of all estimated intervals.

Besides, the values of log-likelihood function (Log-LF) and the corresponding Akaike information criterion (AIC) are adopted to demonstrate the goodness-of-fit. Subsequently, we use the average values of the Log-LF and AIC based on $N$ Monte Carlo simulations to illustrate the performance of the models $M_0$ and $M_1$. The AIC is defined as follows:

$$AIC = -2 \times \ell(\hat{\theta}|\boldsymbol{y}) + 2p, \tag{19}$$

where $\hat{\theta}$ is the MLE of $\boldsymbol{\theta}$, and $p$ is the number of parameters in vector $\boldsymbol{\theta}$.

The Monte Carlo simulations are performed with accelerated degradation data simulated through six scenarios under model $M_0$. For simplicity, we assume that the performance degradation data of each test unit are observed at common times $\boldsymbol{t}_i = (t_{i1}, t_{i2}, ..., t_{im_i})'$ under test group $\boldsymbol{x}_i, i = 1, 2, ..., l$. Therefore, the set $\boldsymbol{m}$ representing the number of measurements can be simplified to $\boldsymbol{m} = (m_1, m_2, ..., m_l)$.

The simulation scenarios are provided as follows. Six different scenarios are used to simulate accelerated degradation data from the proposed model $M_0$, when the number of stress loading levels set at $l = 4$ and $l = 5$, respectively. Under each scenario, 1000 replicates of accelerated degradation data for $n_i$ units were generated under test group $\boldsymbol{x}_i, i = 1, 2, ..., l$, with parameters setting:

$$\mathbf{I} : (\mu_a, \sigma_a^2, \sigma_B^2, \sigma_\varepsilon^2, \gamma_1, \gamma_2, \gamma_3, \gamma_4) = (20, 0.05, 0.10, 0.10, -1.50, 1.25, -1.50, 0.50).$$

The normal operating conditions are set at $\boldsymbol{x}_0 = (2.5, 0.5, 1.35, 0.35)$, and the failure threshold is defined as $\omega = 10$. Subsequently, models $M_0$ and $M_1$ are utilized to fit the simulated accelerated degradation data, respectively. The details of the simulation scenarios are summarized in Table 1.

We calculate the relative-bias and RMSE of MLEs for unknown parameters and reliability metrics under normal operating conditions, based on 1000 Monte Carlo replicates for degradation models $M_0$ and $M_1$. The results are listed in Tables 2–5. Tables 2 and 3 present a comparative analysis results of Relative-bias and RMSE of MLEs for model parameters and some reliability metrics under models $M_0$ and $M_1$; Tables 4 and 5 present a comparative analysis results of CP and AW of 95% BCa bootstrap $p$ CIs for model parameters with 1000 replicates and parameter setting under models $M_0$ and $M_1$, and where $R(60)$ and $R(100)$ denote the reliability at mission time 60 and 100 hours under normal operating conditions, respectively. The $t_{0,0.1}$ and $t_{0,0.9}$ are the 10th, 90th quantile lifetime of products under normal operating conditions, respectively. Besides, to compare the goodness-of-fit between the proposed model $M_0$ and reference model $M_1$, the average values of the Log-LF and AIC are also provided in Table 6.

Tables 2–5 show that the relative-bias and RMSE of unknown parameters and reliability metrics under normal operating conditions from model $M_0$ are much smaller than those from reference model $M_1$. Compared to model $M_0$, the relative-bias and RMSE of the unknown parameters and reliability metrics produced by model $M_1$ are significantly larger. The reason for this is that the measurement errors were ignored when fitting the ADT data from model $M_0$ using model $M_1$, resulting in more larger relative-bias and RMSE. This clearly shows that the measurement errors should be taken into account in ADT modelling. Therefore, in order

**Table 1. The simulation scenarios for Wiener process-based MCSADT model.**

| S/N. | $x$ | $n$ | $m$ | $t$ |
|---|---|---|---|---|
| 1 | $\begin{pmatrix} 1.50 & 0.60 & 1.50 & 0.50 \\ 1.25 & 0.90 & 1.75 & 0.75 \\ 1.00 & 1.20 & 2.00 & 1.00 \\ 0.75 & 1.50 & 2.25 & 1.25 \end{pmatrix}$ | (15,15,10,10) | (8,8,8,8) | (1, 3, 5, 7, 9, 11, 13, 15) <br> (1, 2, 4, 6, 8, 10, 12, 15) <br> (1, 3, 4, 6, 8, 12, 13, 15) <br> (1, 3, 5, 7, 9, 10, 12, 15) |
| 2 | $\begin{pmatrix} 1.50 & 0.60 & 1.50 & 0.50 \\ 1.25 & 0.90 & 1.75 & 0.75 \\ 1.00 & 1.20 & 2.00 & 1.00 \\ 0.75 & 1.50 & 2.25 & 1.25 \end{pmatrix}$ | (20,15,15,10) | (8,8,8,8) | (1, 3, 5, 7, 9, 11, 13, 15) <br> (1, 2, 4, 6, 8, 10, 12, 15) <br> (1, 3, 4, 6, 8, 12, 13, 15) <br> (1, 3, 5, 7, 9, 10, 12, 15) |
| 3 | $\begin{pmatrix} 1.50 & 0.60 & 1.50 & 0.50 \\ 1.25 & 0.90 & 1.75 & 0.75 \\ 1.00 & 1.20 & 2.00 & 1.00 \\ 0.75 & 1.50 & 2.25 & 1.25 \end{pmatrix}$ | (25,20,20,15) | (10,10,10,10) | (1, 3, 5, 7, 9, 11, 13, 15, 16, 18) <br> (1, 2, 4, 6, 8, 10, 12, 14, 16, 18) <br> (1, 3, 4, 6, 8, 10, 13, 15, 17, 18) <br> (1, 3, 5, 7, 9, 10, 12, 14, 16, 18) |
| 4 | $\begin{pmatrix} 1.50 & 0.60 & 1.50 & 0.50 \\ 1.25 & 0.90 & 1.75 & 0.75 \\ 1.00 & 1.20 & 2.00 & 1.00 \\ 0.75 & 1.50 & 2.25 & 1.25 \\ 0.50 & 1.80 & 2.50 & 1.50 \end{pmatrix}$ | (30,25,20,20,15) | (15,12,12,10,10) | (1, 2, 4, 5, 6, 7, 8, 10, 12, 14, 15, 16, 18, 20, 22) <br> (1, 2, 3, 4, 5, 6, 8, 10, 12, 14, 16, 18) <br> (1, 2, 3, 4, 5, 6, 8, 10, 12, 14, 16, 18) <br> (1, 2, 3, 4, 5, 7, 10, 12, 15, 18) <br> (1, 2, 3, 4, 6, 8, 10, 12, 14, 16) |
| 5 | $\begin{pmatrix} 1.50 & 0.60 & 1.50 & 0.50 \\ 1.25 & 0.90 & 1.75 & 0.75 \\ 1.00 & 1.20 & 2.00 & 1.00 \\ 0.75 & 1.50 & 2.25 & 1.25 \\ 0.50 & 1.80 & 2.50 & 1.50 \end{pmatrix}$ | (35,30,25,25,20) | (18,15,15,12,12) | (1, 2, 3, 4, 5, 6, 7, 8, 9, 10, 11, 12, 13, 14, 15, 16, 18, 22) <br> (1, 2, 4, 5, 6, 7, 8, 10, 12, 14, 15, 16, 18, 20, 22) <br> (1, 2, 4, 5, 6, 7, 8, 10, 12, 14, 15, 16, 18, 20, 22) <br> (1, 2, 3, 4, 5, 6, 8, 10, 12, 14, 16, 18) <br> (1, 2, 3, 4, 5, 6, 8, 10, 12, 14, 16, 18) |
| 6 | $\begin{pmatrix} 1.50 & 0.60 & 1.50 & 0.50 \\ 1.25 & 0.90 & 1.75 & 0.75 \\ 1.00 & 1.20 & 2.00 & 1.00 \\ 0.75 & 1.50 & 2.25 & 1.25 \\ 0.50 & 1.80 & 2.50 & 1.50 \end{pmatrix}$ | (40,35,30,30,25) | (20,18,18,15,15) | (1, 2, 3, 4, 5, 6, 7, 8, 9, 10, 11, 12, 13, 14, 15, 16, 18, 19, 20, 22) <br> (1, 2, 3, 4, 5, 6, 7, 8, 9, 10, 11, 12, 13, 14, 15, 16, 18, 22) <br> (1, 2, 3, 4, 5, 6, 7, 8, 9, 10, 11, 12, 13, 14, 15, 16, 18, 22) <br> (1, 2, 3, 4, 5, 6, 7, 8, 9, 10, 11, 12, 13, 14, 15) <br> (1, 2, 3, 4, 5, 6, 7, 8, 9, 10, 11, 12, 13, 14, 15) |

**Table 2. Relative-bias and RMSE of MLEs for model parameters and some reliability metrics under model $M_0$.**

| **Relative-bias** | | | | | | | | | | | | | |
|---|---|---|---|---|---|---|---|---|---|---|---|---|---|
| **S/N.** | **Unknown parameters** | | | | | | | | **Reliability metrics** | | | | |
| | $\mu_a$ | $\sigma_a^2$ | $\sigma_B^2$ | $\sigma_\varepsilon^2$ | $\gamma_1$ | $\gamma_2$ | $\gamma_3$ | $\gamma_4$ | $t_{0,MTTF}$ | $R(60)$ | $R(100)$ | $t_{0,0.1}$ | $t_{0,0.5}$ | $t_{0,0.9}$ |
| 1 | 0.1567 | 0.0980 | 0.0516 | 0.1230 | -0.0226 | -0.0087 | -0.0236 | 0.0969 | 0.0243 | 0.0062 | 0.1820 | 0.0347 | 0.0215 | 0.0094 |
| 2 | -0.1751 | 0.0976 | 0.0559 | 0.1065 | -0.0232 | 0.0194 | -0.0402 | 0.0829 | 0.0200 | -0.0057 | 0.1450 | 0.0314 | 0.0170 | -0.0066 |
| 3 | -0.1302 | 0.1623 | 0.0996 | 0.1907 | -0.0196 | 0.0070 | -0.0311 | 0.1032 | 0.0236 | 0.0050 | 0.1200 | 0.0316 | 0.0187 | -0.0059 |
| 4 | -0.2518 | 0.2154 | 0.2300 | 0.1357 | -0.0495 | -0.0286 | -0.0947 | -0.0082 | -0.0410 | -0.1079 | -0.0923 | -0.0188 | -0.0469 | -0.0733 |
| 5 | -0.2057 | 0.2584 | 0.2381 | 0.2345 | -0.0397 | -0.0158 | -0.0615 | 0.0174 | -0.0319 | -0.0916 | -0.0327 | -0.0074 | -0.0385 | -0.0678 |
| 6 | -0.1942 | 0.2566 | 0.1898 | 0.2810 | -0.0390 | -0.0114 | -0.0536 | 0.0202 | -0.0328 | -0.0868 | -0.0746 | -0.0139 | -0.0378 | -0.0605 |
| **RMSE** | | | | | | | | | | | | | |
| **S/N.** | **Unknown parameters** | | | | | | | | **Reliability metrics** | | | | |
| | $\mu_a$ | $\sigma_a^2$ | $\sigma_B^2$ | $\sigma_\varepsilon^2$ | $\gamma_1$ | $\gamma_2$ | $\gamma_3$ | $\gamma_4$ | $t_{0,MTTF}$ | $R(60)$ | $R(100)$ | $t_{0,0.1}$ | $t_{0,0.5}$ | $t_{0,0.9}$ |
| 1 | 0.1421 | 0.0153 | 0.0139 | 0.0297 | 0.0030 | 0.0333 | 0.0629 | 0.2222 | 0.0096 | 0.0097 | 0.2475 | 0.0096 | 0.0072 | 0.0059 |
| 2 | 0.0910 | 0.0143 | 0.0134 | 0.0225 | 0.0027 | 0.0327 | 0.0412 | 0.2460 | 0.0083 | 0.0106 | 0.1931 | 0.0086 | 0.0060 | 0.0053 |
| 3 | 0.0782 | 0.0307 | 0.0188 | 0.0492 | 0.0023 | 0.0116 | 0.0256 | 0.2301 | 0.0065 | 0.0071 | 0.1667 | 0.0060 | 0.0059 | 0.0044 |
| 4 | 0.0530 | 0.0494 | 0.0603 | 0.0275 | 0.0035 | 0.0032 | 0.0188 | 0.0224 | 0.0028 | 0.0161 | 0.0497 | 0.0018 | 0.0033 | 0.0064 |
| 5 | 0.0460 | 0.0698 | 0.0608 | 0.0420 | 0.0017 | 0.0015 | 0.0058 | 0.0143 | 0.0019 | 0.0094 | 0.0179 | 0.0009 | 0.0018 | 0.0049 |
| 6 | 0.0400 | 0.0677 | 0.0401 | 0.0482 | 0.0017 | 0.009 | 0.0031 | 0.0120 | 0.0014 | 0.0085 | 0.0289 | 0.0008 | 0.0016 | 0.0039 |

**Table 3. Relative-bias and RMSE of MLEs for model parameters and some reliability metrics under model $M_1$.**

**Relative-bias**

| S/N. | Unknown parameters | | | | | | | | Reliability metrics | | | | | |
|---|---|---|---|---|---|---|---|---|---|---|---|---|---|---|
| | $\mu_a$ | $\sigma_a^2$ | $\sigma_B^2$ | $\sigma_\varepsilon^2$ | $\gamma_1$ | $\gamma_2$ | $\gamma_3$ | $\gamma_4$ | $t_{0,MTTF}$ | $R(60)$ | $R(100)$ | $t_{0,0.1}$ | $t_{0,0.5}$ | $t_{0,0.9}$ |
| 1 | 0.6107 | 0.6570 | 0.7356 | - | 0.1239 | 0.1700 | 0.1519 | 0.3089 | 0.1641 | 0.4546 | 0.4880 | -0.2845 | -0.5545 | 0.3659 |
| 2 | 0.5910 | 0.5450 | 0.6700 | - | 0.1345 | 0.1695 | 0.1348 | 0.2687 | 0.1191 | 0.3995 | 0.4347 | 0.2375 | 0.5179 | 0.3495 |
| 3 | 0.5799 | 0.5005 | 0.5927 | - | -0.1245 | -0.1881 | -0.1620 | 0.2237 | -0.1167 | -0.2792 | -0.3956 | -0.2542 | -0.5097 | 0.3115 |
| 4 | -0.4315 | 0.5107 | 1.8217 | - | 0.0747 | -0.0174 | -0.1173 | 0.1554 | 0.0913 | -0.1909 | 0.3353 | -0.1745 | 0.0597 | 0.2459 |
| 5 | 0.4429 | 0.4437 | 2.1215 | - | 0.0765 | -0.0196 | -0.1054 | 0.1442 | -0.0866 | 0.1860 | 0.3226 | 0.2355 | 0.0463 | 0.2555 |
| 6 | 0.4105 | 0.3000 | 2.6364 | - | 0.0659 | 0.0114 | 0.1357 | -0.1400 | 0.0823 | -0.1745 | 0.3146 | 0.2549 | 0.0159 | 0.1798 |

**RMSE**

| S/N. | Unknown parameters | | | | | | | | Reliability metrics | | | | | |
|---|---|---|---|---|---|---|---|---|---|---|---|---|---|---|
| | $\mu_a$ | $\sigma_a^2$ | $\sigma_B^2$ | $\sigma_\varepsilon^2$ | $\gamma_1$ | $\gamma_2$ | $\gamma_3$ | $\gamma_4$ | $t_{0,MTTF}$ | $R(60)$ | $R(100)$ | $t_{0,0.1}$ | $t_{0,0.5}$ | $t_{0,0.9}$ |
| 1 | 0.6447 | 0.8718 | 0.6694 | - | 0.0287 | 0.0530 | 0.1070 | 0.6265 | 0.0231 | 0.4251 | 0.4719 | 0.1132 | 0.0610 | 0.0715 |
| 2 | 0.5678 | 0.8102 | 0.6929 | - | 0.0275 | 0.0455 | 0.0986 | 0.6027 | 0.0189 | 0.3766 | 0.4457 | 0.0973 | 0.0533 | 0.0718 |
| 3 | 0.5111 | 0.7445 | 0.5956 | - | 0.0197 | 0.0411 | 0.0893 | 0.6069 | 0.0121 | 0.1226 | 0.4383 | 0.0691 | 0.0526 | 0.0659 |
| 4 | 0.4133 | 0.4671 | 0.4364 | - | 0.0119 | 0.0128 | 0.0642 | 0.2140 | 0.0119 | 0.0385 | 0.3644 | 0.0364 | 0.0256 | 0.0588 |
| 5 | 0.3954 | 0.4349 | 0.5144 | - | 0.0151 | 0.0156 | 0.0695 | 0.2265 | 0.0104 | 0.0396 | 0.3261 | 0.0463 | 0.0185 | 0.0671 |
| 6 | 0.4218 | 0.3886 | 0.4703 | - | 0.0142 | 0.0108 | 0.0598 | 0.2102 | 0.0105 | 0.0337 | 0.2361 | 0.0397 | 0.0159 | 0.0598 |

**Table 4. CP and AW of 95% BCa bootstrap $p$ CIs for model parameters with 1000 repliciates and parameter setting I ($M_0$).**

**CP**

| S/N. | Unknown parameters | | | | | | | | Reliability metrics | | | | | |
|---|---|---|---|---|---|---|---|---|---|---|---|---|---|---|
| | $\mu_a$ | $\sigma_a^2$ | $\sigma_B^2$ | $\sigma_\varepsilon^2$ | $\gamma_1$ | $\gamma_2$ | $\gamma_3$ | $\gamma_4$ | $t_{0,MTTF}$ | $R(60)$ | $R(100)$ | $t_{0,0.1}$ | $t_{0,0.5}$ | $t_{0,0.9}$ |
| 1 | 0.9333 | 0.9382 | 0.9412 | 0.9363 | 0.9445 | 0.9427 | 0.9383 | 0.9355 | 0.9362 | 0.9125 | 0.9289 | 0.9294 | 0.9337 | 0.9405 |
| 2 | 0.9417 | 0.9415 | 0.9420 | 0.9366 | 0.9385 | 0.9401 | 0.9406 | 0.9423 | 0.9383 | 0.9174 | 0.9296 | 0.9340 | 0.9357 | 0.9425 |
| 3 | 0.9469 | 0.9470 | 0.9458 | 0.9418 | 0.9397 | 0.9381 | 0.9422 | 0.9397 | 0.9417 | 0.9195 | 0.9338 | 0.9368 | 0.9398 | 0.9438 |
| 4 | 0.9490 | 0.9505 | 0.9471 | 0.9437 | 0.9436 | 0.9429 | 0.9458 | 0.9466 | 0.9437 | 0.9216 | 0.9381 | 0.9404 | 0.9426 | 0.9456 |
| 5 | 0.9509 | 0.9496 | 0.9498 | 0.9475 | 0.9483 | 0.9475 | 0.9487 | 0.9480 | 0.9453 | 0.9256 | 0.9421 | 0.9456 | 0.9450 | 0.9497 |
| 6 | 0.9515 | 0.9514 | 0.9520 | 0.9508 | 0.9517 | 0.9500 | 0.9509 | 0.9506 | 0.9477 | 0.9335 | 0.9456 | 0.9484 | 0.9473 | 0.9510 |

**AW**

| S/N. | Unknown parameters | | | | | | | | Reliability metrics | | | | | |
|---|---|---|---|---|---|---|---|---|---|---|---|---|---|---|
| | $\mu_a$ | $\sigma_a^2$ | $\sigma_B^2$ | $\sigma_\varepsilon^2$ | $\gamma_1$ | $\gamma_2$ | $\gamma_3$ | $\gamma_4$ | $t_{0,MTTF}$ | $R(60)$ | $R(100)$ | $t_{0,0.1}$ | $t_{0,0.5}$ | $t_{0,0.9}$ |
| 1 | 48.881 | 0.0158 | 0.0484 | 0.0644 | 0.5208 | 1.0601 | 1.5436 | 1.3245 | 38.865 | 0.4521 | 0.4584 | 22.439 | 36.316 | 59.741 |
| 2 | 34.021 | 0.0131 | 0.0420 | 0.0565 | 0.4607 | 0.9400 | 1.2532 | 1.1797 | 36.894 | 0.4053 | 0.4154 | 20.475 | 32.775 | 48.924 |
| 3 | 33.436 | 0.0129 | 0.0393 | 0.0525 | 0.4530 | 0.8611 | 1.2117 | 1.0996 | 35.172 | 0.3958 | 0.3952 | 19.982 | 30.824 | 45.978 |
| 4 | 7.082 | 0.0092 | 0.0254 | 0.0249 | 0.2144 | 0.5674 | 0.5796 | 0.6534 | 13.465 | 0.2765 | 0.0909 | 7.788 | 12.597 | 20.779 |
| 5 | 6.581 | 0.0081 | 0.0238 | 0.0224 | 0.1889 | 0.4889 | 0.5218 | 0.5742 | 12.152 | 0.2422 | 0.0860 | 6.736 | 11.271 | 19.232 |
| 6 | 5.685 | 0.0073 | 0.0213 | 0.0218 | 0.1687 | 0.4169 | 0.4845 | 0.4246 | 10.987 | 0.2225 | 0.0763 | 5.842 | 10.286 | 17.186 |

to obtain effective evaluation results, we should carefully consider a more appropriate degradation model.

Table 6 clearly shows that the proposed model $M_0$ performs better than the reference model $M_1$ in terms of Log-LF and AIC for each simulation scenario. It can be concluded that the proposed Wiener process MCSADT model with three-source variability performs better than some existing simple degradation models in literature.

## 4.2 Numerical example

In this section, an illustrative example is analysed to investigate the performance of the proposed method. To evaluate the reliability of a particular product under normal operating conditions, a MCSADT with four acceleration variables was performed in a laboratory. In this

**Table 5. CP and AW of 95% BCa bootstrap $p$ CIs for model parameters with 1000 repliciates and parameter setting I ($M_1$).**

| CP | | | | | | | | | | | | | | |
|---|---|---|---|---|---|---|---|---|---|---|---|---|---|---|
| S/N. | Unknown parameters | | | | | | | | Reliability metrics | | | | | |
| | $\mu_a$ | $\sigma_a^2$ | $\sigma_B^2$ | $\sigma_\varepsilon^2$ | $\gamma_1$ | $\gamma_2$ | $\gamma_3$ | $\gamma_4$ | $t_{0,MTTF}$ | $R(60)$ | $R(100)$ | $t_{0,0.1}$ | $t_{0,0.5}$ | $t_{0,0.9}$ |
| 1 | 0.8997 | 0.8892 | 0.8892 | - | 0.8765 | 0.8485 | 0.8789 | 0.8647 | 0.8892 | 0.8825 | 0.8758 | 0.8792 | 0.8748 | 0.8801 |
| 2 | 0.9030 | 0.9053 | 0.8938 | - | 0.8859 | 0.8661 | 0.8996 | 0.8718 | 0.8898 | 0.8893 | 0.8843 | 0.8809 | 0.8865 | 0.8834 |
| 3 | 0.8982 | 0.9065 | 0.9056 | - | 0.8982 | 0.8826 | 0.9025 | 0.8819 | 0.8994 | 0.8916 | 0.8935 | 0.8963 | 0.8890 | 0.8969 |
| 4 | 0.9110 | 0.9154 | 0.9197 | - | 0.9102 | 0.8931 | 0.9055 | 0.8975 | 0.9071 | 0.8923 | 0.9013 | 0.9100 | 0.8954 | 0.9036 |
| 5 | 0.9194 | 0.9109 | 0.9095 | - | 0.9093 | 0.9066 | 0.9146 | 0.9056 | 0.9059 | 0.8999 | 0.9091 | 0.9074 | 0.9052 | 0.9191 |
| 6 | 0.9209 | 0.9168 | 0.9212 | - | 0.9082 | 0.9079 | 0.9005 | 0.9103 | 0.9136 | 0.9064 | 0.9137 | 0.9195 | 0.9139 | 0.9079 |
| AW | | | | | | | | | | | | | | |
| S/N. | Unknown parameters | | | | | | | | Reliability metrics | | | | | |
| | $\mu_a$ | $\sigma_a^2$ | $\sigma_B^2$ | $\sigma_\varepsilon^2$ | $\gamma_1$ | $\gamma_2$ | $\gamma_3$ | $\gamma_4$ | $t_{0,MTTF}$ | $R(60)$ | $R(100)$ | $t_{0,0.1}$ | $t_{0,0.5}$ | $t_{0,0.9}$ |
| 1 | 58.736 | 0.0235 | 0.0736 | - | 0.7658 | 1.8861 | 2.2154 | 1.9465 | 50.337 | 0.8291 | 0.8864 | 38.956 | 48.658 | 74.211 |
| 2 | 49.911 | 0.0219 | 0.0698 | - | 0.7015 | 1.7650 | 2.1122 | 1.7783 | 44.384 | 0.7483 | 0.7995 | 32.575 | 42.185 | 69.289 |
| 3 | 43.356 | 0.0193 | 0.0628 | - | 0.6558 | 1.6638 | 1.8798 | 1.5960 | 38.892 | 0.6398 | 0.6926 | 28.379 | 38.484 | 61.169 |
| 4 | 26.429 | 0.0155 | 0.0443 | - | 0.4094 | 1.2637 | 1.3976 | 1.1542 | 20.119 | 0.4475 | 0.2899 | 15.283 | 22.127 | 39.258 |
| 5 | 21.771 | 0.0140 | 0.0386 | - | 0.3425 | 1.0989 | 1.1698 | 0.8755 | 17.472 | 0.3830 | 0.2384 | 12.558 | 18.944 | 32.952 |
| 6 | 15.998 | 0.0127 | 0.0349 | - | 0.2989 | 0.8559 | 1.0872 | 0.7026 | 15.309 | 0.3524 | 0.1782 | 10.347 | 16.773 | 28.415 |

**Table 6. Average Log-LF and AIC for MCSADT models $M_0$ and $M_1$.**

| Model | Log-LF | | | | | | AIC | | | | | |
|---|---|---|---|---|---|---|---|---|---|---|---|---|
| | CS 1 | CS 2 | CS 3 | CS 4 | CS 5 | CS 6 | CS 1 | CS 2 | CS 3 | CS 4 | CS 5 | CS 6 |
| $M_0$ | -365.96 | -440.08 | -757.40 | -1232.79 | -1827.82 | -2453.16 | 381.96 | 456.08 | 773.40 | 1248.79 | 1843.82 | 2469.16 |
| $M_1$ | -540.50 | -755.28 | -1211.42 | -1302.86 | -1939.05 | -2675.06 | 556.50 | 769.28 | 1227.42 | 1318.86 | 1955.05 | 2691.06 |

**Table 7. The test conditions for a MCSADT with four acceleration variables and four stress loading levels.**

| $x$ | $n$ | $t$ |
|---|---|---|
| $\begin{pmatrix} 1.50 & 0.60 & 1.50 & 0.50 \\ 1.25 & 0.90 & 1.75 & 0.75 \\ 1.00 & 1.20 & 2.00 & 1.00 \\ 0.75 & 1.50 & 2.25 & 1.25 \end{pmatrix}$ | $(15,15,10,10)$ | $(1,2,3,4,6,8,10,12,14,16,19,22,24)$ <br> $(1,2,3,4,5,7,9,11,13,16,19,22,24)$ <br> $(1,2,3,4,5,7,10,12,15,18,20,22,24)$ <br> $(1,2,3,4,6,8,10,12,14,16,18,20,24)$ |

MCSADT, the test conditions, including the stress loading level, sample size, and measurement time, are listed in Table 7. In addition, the original data are shown in Fig 1. The failure threshold for the normal stress loading level is specified as $\omega = 10$, and the normal operating conditions was set as $x_0 = (2.5, 0.5, 1.35, 0.35)$. The degradation model $M_1$ is utilized as a reference model for comparison purposes.

To illustrate the superiority of the proposed model $M_0$, the degradation models $M_0$ and $M_1$ are used to analyze the degradation dataset shown in Fig 1. To show the goodness-of-fit of the two degradation models, the corresponding log-LF and AIC values are also calculated. The MLEs and confidence intervals of the model parameters and some reliability metrics under normal operating conditions, as well as the values of Log-LF and AIC for the two degradation models are determined. The corresponding results are shown in Tables 8 and 9. Table 8 shows that the degradation model $M_0$ has a better fit for both the log-LF and AIC values.

Based on the MLEs of the model parameters shown in Table 9, it is not difficult to determine the estimated mean degradation paths for models $M_0$ and $M_1$, which are shown in Fig 2. Fig 3 shows that the degradation model $M_0$ is superior to the model $M_1$, as the data set originates from model $M_0$ and not from model $M_1$. Besides, the reliability at mission time is

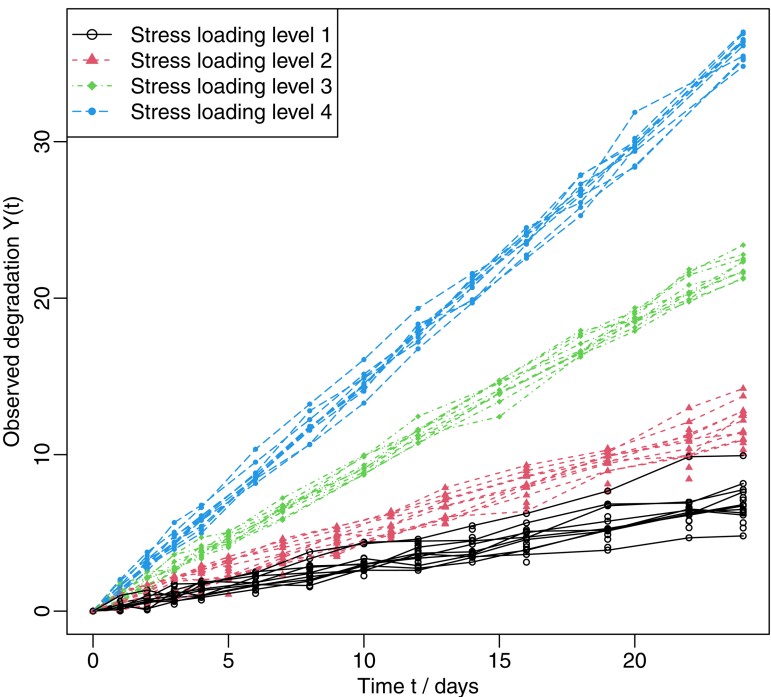

**Fig 1. Observed degradation under four accelerated stress loading levels.**

**Table 8. MLEs of unknown parameters and reliability metrics for degradation models $M_0$ and $M_1$.**

| Model | Unknown parameters | | | | | | | | Reliability metrics | | | | | Log-LF | AIC |
|---|---|---|---|---|---|---|---|---|---|---|---|---|---|---|---|
| | $\mu_a$ | $\sigma_a^2$ | $\sigma_B^2$ | $\sigma_\varepsilon^2$ | $\gamma_1$ | $\gamma_2$ | $\gamma_3$ | $b$ | $t_{0,MTTF}$ | $R(100)$ | $t_{0,0.1}$ | $t_{0,0.5}$ | $t_{0,0.9}$ | | |
| $M_0$ | 10.280 | 0.030 | 0.056 | 0.067 | -1.490 | 1.450 | -1.670 | 0.630 | 149.656 | 0.898 | 207.351 | 143.673 | 99.766 | -408.637 | 424.637 |
| $M_1$ | 13.530 | 0.060 | 0.140 | - | -1.520 | 1.530 | -1.880 | 0.710 | 150.836 | 0.774 | 242.065 | 136.616 | 77.871 | -450.166 | 464.166 |

**Table 9. BCa bootstrap confidence intervals for unknown parameters and reliability metrics ( $1 - \alpha = 0.95$).**

| Model | Unknown parameters | | | | | | | |
|---|---|---|---|---|---|---|---|---|
| | $\mu_a$ | $\sigma_a^2$ | $\sigma_B^2$ | $\sigma_\varepsilon^2$ | $\gamma_1$ | $\gamma_2$ | $\gamma_3$ | $\gamma_4$ |
| $M_0$ | [6.73, 16.83] | [0.023, 0.033] | [0.043, 0.064] | [0.046, 0.082] | [-1.600, -1.329] | [1.279, 1.650] | [-1.942, -1.520] | [0.619, 0.961] |
| $M_1$ | [4.58, 20.76] | [0.008, 0.126] | [0.007, 0.151] | - | [-2.529, -0.635] | [0.065, 2.983] | [-3.099, -0.451] | [0.438, 1.426] |
| **Model** | **Reliability metrics** | | | | | | | |
| | $R(90)$ | $R(120)$ | $R(150)$ | $R(180)$ | $t_{0,MTTF}$ | $t_{0,0.1}$ | $t_{0,0.5}$ | $t_{0,0.9}$ |
| $M_0$ | [0.889, 0.984] | [0.549, 0.821] | [0.228, 0.565] | [0.074, 0.343] | [128.082, 166.163] | [172.902, 239.161] | [123.989, 157.466] | [88.503, 109.875] |
| $M_1$ | [0.768, 0.995] | [0.408, 0.926] | [0.137, 0.710] | [0.026, 0.489] | [104.174, 198.385] | [135.458, 268.930] | [101.362, 179.445] | [65.397, 147.568] |

calculated based on the estimated models $M_0$ and $M_1$, and the reliability curves for the two degradation models are shown in Fig 3. From Fig 3, it can be concluded that the reliability of model $M_0$ is higher than that of model $M_1$ for a duration of use $t \leq 158$ days. Over time, when the operating time $t$ approximate greater 158 days, the reliability of model $M_0$ decreases more than that of model $M_1$.

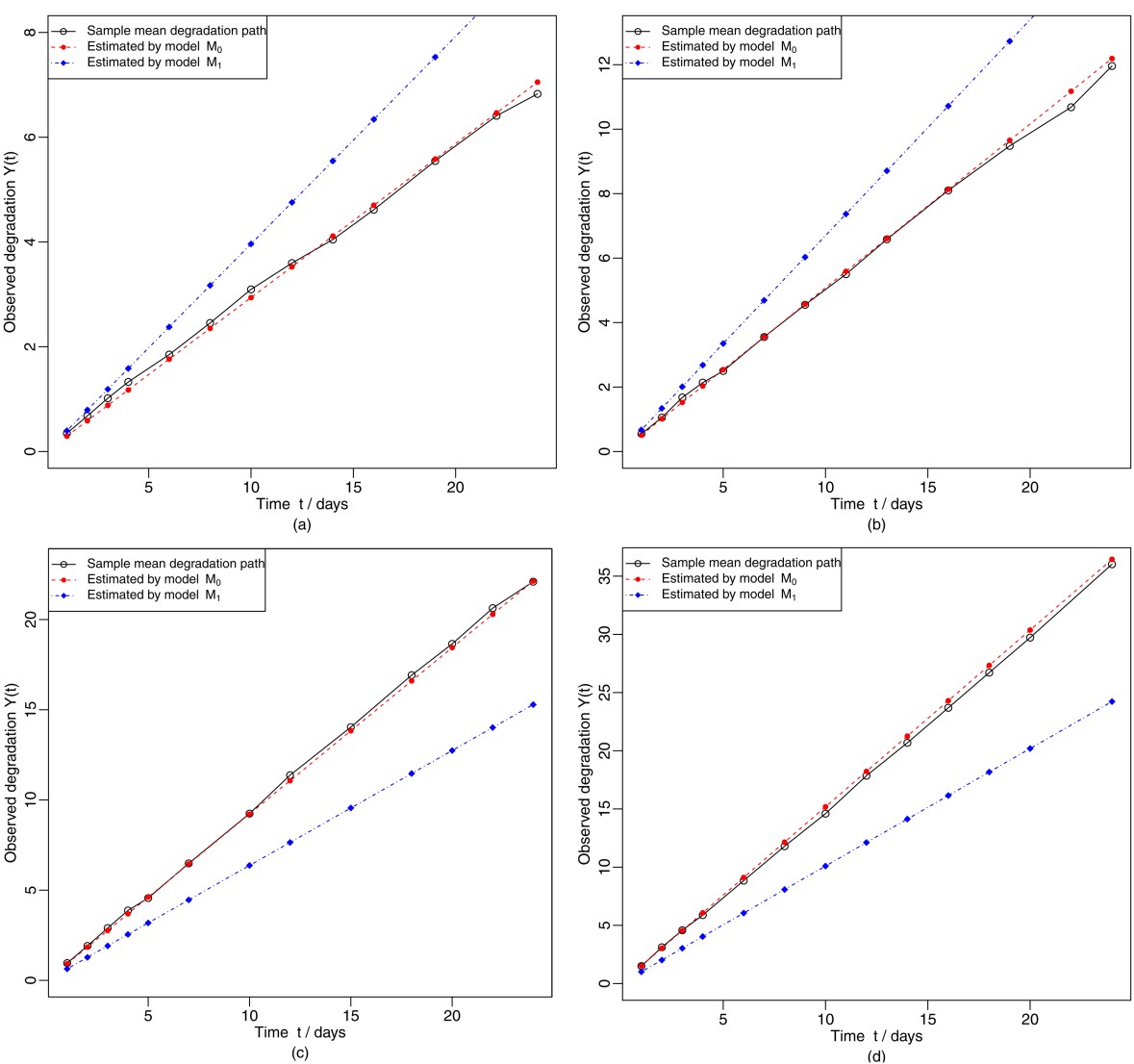

**Fig 2. Estimated mean degradation paths for units using degradation models $M_0$ and $M_1$.**

## Conclusion

In this article, we investigate a reliability assessment model for linear Wiener accelerated degradation model with multiple accelerating variables, which simultaneously accounts for temporal variability, unit-to-unit variability, and measurement variability. The proposed accelerated degradation model more accurately reflects real-world conditions and improves the realism and robustness of degradation analysis by incorporating random effects and measurement variability. The explicit expression of the lifetime distribution for the proposed linear Wiener multiple constant-stress accelerated degradation model under normal operating conditions is derived. In addition, the maximum likelihood estimates of the model parameters are derived using the profile likelihood approach, and the maximum likelihood estimates of certain reliability metrics under normal operating conditions are obtained through the invariance property of the maximum likelihood method. The confidence intervals for the

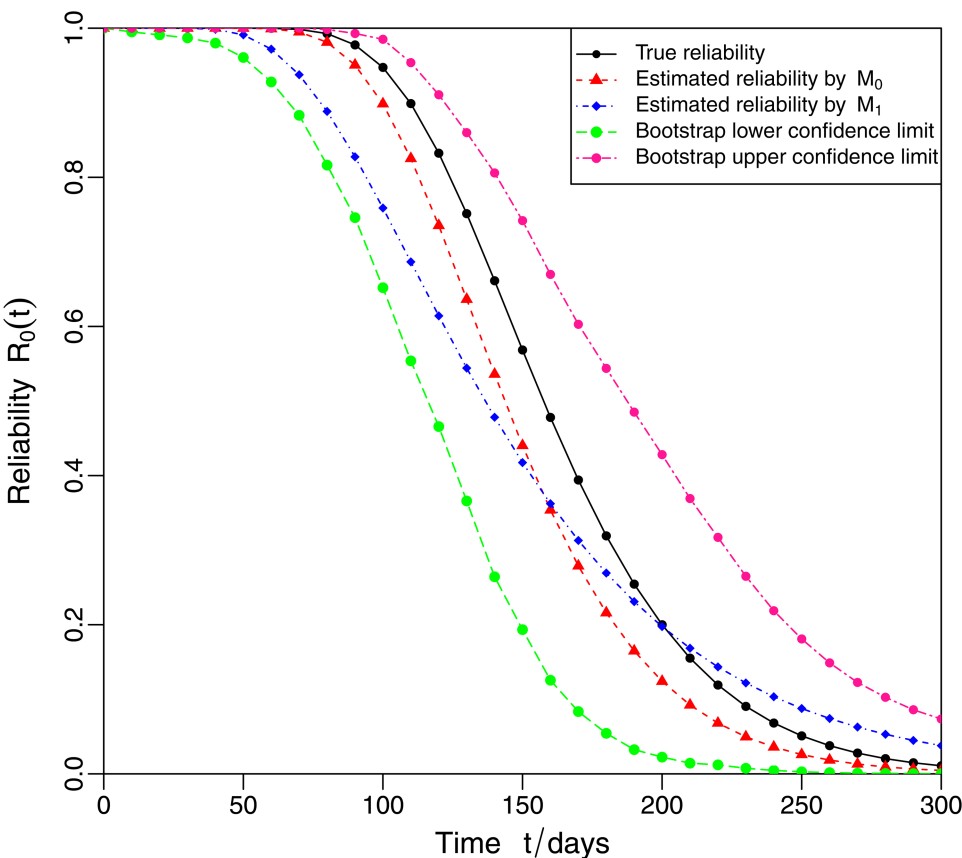

**Fig 3. The reliability curve $R_0(t)$ under normal operating conditions $x_0$.**

model parameters and reliability metrics are constructed using the bias-corrected and accelerated percentile method, which provides more accurate inference, particularly in small-sample scenarios. The performance of the proposed procedure was guaranteed based on extensive simulations and a numerical example.

In future research, nonlinear accelerated degradation models incorporating random effects and initial degradation levels will be of significant interest. A time scale transformation function with unknown parameters may be integrated into the accelerated degradation model to better capture nonlinear degradation patterns.

## Acknowledgments

The authors are deeply grateful to the editor, the associate editor, and the reviewers for their insightful comments and constructive suggestions for this paper.

## Author contributions

**Conceptualization:** Qianqian Huang, JIAYIN TANG.

**Data curation:** Qianqian Huang, JIAYIN TANG, Xuefeng Feng.

**Formal analysis:** Qianqian Huang, Xuefeng Feng.

**Funding acquisition:** Qianqian Huang, JIAYIN TANG.

**Investigation:** Qianqian Huang, Xuefeng Feng.

**Methodology:** Qianqian Huang, JIAYIN TANG.

**Resources:** Qianqian Huang, JIAYIN TANG.

**Supervision:** Qianqian Huang, JIAYIN TANG.

**Writing – original draft:** Xuefeng Feng.

**Writing – review & editing:** Xuefeng Feng.

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
