## [Decision Letter · Decision Letter 0]

21 Jan 2025

PONE-D-24-52301Reliability assessment model for multiple stress factors accelerated degradation test using a Wiener process with random effectsPLOS ONE

Dear Dr. TANG,

Thank you for submitting your manuscript to PLOS ONE. After careful consideration, we feel that it has merit but does not fully meet PLOS ONE’s publication criteria as it currently stands. Therefore, we invite you to submit a revised version of the manuscript that addresses the points raised during the review process.

We look forward to receiving your revised manuscript.

Kind regards,

Qingan Qiu

Academic Editor

PLOS ONE

Journal Requirements:

2. In the online submission form you indicate that your data is not available for proprietary reasons and have provided a contact point for accessing this data. Please note that your current contact point is a co-author on this manuscript. According to our Data Policy, the contact point must not be an author on the manuscript and must be an institutional contact, ideally not an individual. Please revise your data statement to a non-author institutional point of contact, such as a data access or ethics committee, and send this to us via return email. Please also include contact information for the third party organization, and please include the full citation of where the data can be found.

5. Please update your submission to use the PLOS LaTeX template. The template and more information on our requirements for LaTeX submissions can be found at http://journals.plos.org/plosone/s/latex.

“This work was supported by the National Social Science Fund of China (Grant No. 23BTJ010).”

“National Social Science Fund of China (Grant No.23BTJ010)”

Additional Editor Comments (if provided):

Five reviewers provide important comments. While reviewers find value in the methodology, they have comments in terms of the statement of contribution, parameter validation and application illustration. A MAJOR REVISION is recommended for a second round of review. The authors should carefully incorporate all comments into revision, particularly paying attention to the contribution and application study.

Reviewers' comments:

Reviewer's Responses to Questions

**Comments to the Author**

1. Is the manuscript technically sound, and do the data support the conclusions?

Reviewer #1: Yes

Reviewer #2: Partly

Reviewer #3: Yes

Reviewer #4: Partly

Reviewer #5: Yes

2. Has the statistical analysis been performed appropriately and rigorously? 

Reviewer #1: Yes

Reviewer #2: Yes

Reviewer #3: Yes

Reviewer #4: Yes

Reviewer #5: Yes

3. Have the authors made all data underlying the findings in their manuscript fully available?

Reviewer #1: Yes

Reviewer #2: Yes

Reviewer #3: Yes

Reviewer #4: No

Reviewer #5: Yes

4. Is the manuscript presented in an intelligible fashion and written in standard English?

Reviewer #1: Yes

Reviewer #2: Yes

Reviewer #3: Yes

Reviewer #4: No

Reviewer #5: Yes

5. Review Comments to the Author

Reviewer #1: The paper mainly proposes an accelerated degradation testing model based on the Wiener process for assessing reliability involving multiple stress factors. The model simultaneously considers multiple stress factors, random effects, and measurement errors, deriving an explicit expression for the lifetime distribution and its approximate mean lifetime under normal operating conditions. In addition, the paper derives maximum likelihood estimates for model parameters and some reliability metrics under normal operating conditions. Meanwhile, confidence intervals for model parameters and some reliability metrics are constructed. Overall, the paper makes certain contributions but still requires significant improvements. Specific comments are as follows:

1) Given the complexity of the model parameters, it is suggested that the authors explore the feasibility of other estimation methods (such as Bayesian methods) and compare the advantages and disadvantages of different methods.

2) The model mentioned in the paper is specific to the linear Wiener process. It is recommended that the authors consider extending the model to the nonlinear case to handle more complex degradation patterns.

3) When constructing confidence intervals, the authors adopted the Bootstrap method. It is suggested that the authors discuss other methods for constructing confidence intervals (such as asymptotic methods) and compare the accuracy and efficiency of different methods.

Reviewer #2: The paper is well written. However, the following concerns should be addressed.

The authors should compare the proposed models with other accelerated degradation models with multiple stress factors.

Reviewer #3: This paper proposes an accelerated degradation test (ADT) model based on a linear Wiener process. This model considers multiple stress factors, random effects and measurement errors, and derives the explicit expression of the life distribution under normal working conditions and the maximum likelihood estimation method of its parameters. This is a good idea. The following are some of my concerns about publishing this paper.

1. The main contribution of this paper is not prominently marked. Please summarize the innovations in the Introduction.

2. How are the parameters of the Wiener process determined and verified?

3. There are many models of random processes. Why is the Wiener process suitable for this case?

4. Recent reliability assessment related work could be referred to, Artificial intelligence enhanced fault prediction with industrial incomplete information, Data-model-linked remaining useful life prediction method with small sample data: A case of subsea valve, Remaining useful life estimation of structure systems under the influence of multiple causes: Subsea pipelines as a case study, Remaining useful life re-prediction methodology based on Wiener process: Subsea Christmas tree system as a case study.

5. I think there is still a lot of room for improvement in the conclusion part, especially the summary of the work done needs to be further strengthened.

Reviewer #4: The paper studied the degradation model using the Wiener process considering multiple stress factors. The topic is interesting. However for the current version, the innovtion is not enough. The authors should revise significantly.

1, the degradation model is too usual without much new contribution, a more realistic and complex models should be considered and the engineering background is needed. The engineering mechanism for the failure process is very important for reliability research. Thus, a novel degardation is needed first and the parameter estamition process could be much brief. For example�considering the shock process, multiple degardation indexes, etc.

2. what is the application value of the research and the results. A more management view is needed for the conclusion.

Reviewer #5: The manuscript studies multiple stress factors accelerated degradation test using a Wiener process with random effects and MLE of model parameters are derived using profile likelihood approach. The manuscript is well-written. Some revisions are suggested as follows:

(1) More discussions should be provided to prove creativity of the present work in the Introduction part.

(2) A nomenclature is suggested since there are too many abbreviations used.

(3) Reasonability and generality of the basic assumptions should be clearly discussed, especial A2 and A3.

(4) Transformed variables x_iq in A3 was referred to as accelerating variables in the beginning of Section 2.2, please have a check and uniform them.

(5) “,” is suggested for notations with subscripts, for example, “t_ijk” shoud be “t_i,j,k” since i, j, k might take values more than 10.

(6) There are many formulae and numbers in this manuscript, please check them carefully to make sure they are all correct without any typos and mistakes.

(7) In the reference list, please list all authors instead of using “et al.”, and the using of spaces should also be carefully uniformed.

6. PLOS authors have the option to publish the peer review history of their article (what does this mean?). If published, this will include your full peer review and any attached files.

Reviewer #1: **Yes: **Rui Peng

Reviewer #2: No

Reviewer #3: No

Reviewer #4: No

Reviewer #5: No

---

## [Author Response · Author response to Decision Letter 1]

7 Mar 2025

Response to Reviewer Comments on PONE-D-24-52301

Reliability assessment model for multiple stress factors accelerated degradation test using a Wiener process with random effects

Jiayin Tang3

3 Department of Statistics, School of Mathematics, Southwest Jiaotong University, Chengdu, Sichuan, China

Dear Editor and Reviewers:

Thank you for your letter and for the insightful comments provided by the reviewer regarding our manuscript (PONE-D-24-52301). These comments are invaluable and greatly assist in enhancing the quality of our manuscript. We have thoroughly considered all the comments and have made corresponding corrections for each. We have carefully considered all of the feedback, our responses to the reviewers' comments are outlined below:

Response to Reviewer 1

Comment 1: Given the complexity of the model parameters, it is suggested that the authors explore the feasibility of other estimation methods (such as Bayesian methods) and compare the advantages and disadvantages of different methods.

Response 1: We sincerely thank the reviewer for the valuable suggestion to explore alternative estimation methods, such as Bayesian approaches, and to compare their respective advantages and disadvantages. In our study, we selected the current estimation method for its computational efficiency and proven effectiveness in addressing the specific characteristics of our data. This method has been widely utilized in similar contexts and has yielded reliable results in our preliminary analyses.

We acknowledge that Bayesian methods could offer significant advantages, particularly in terms of incorporating prior knowledge and addressing uncertainty. However, their implementation may necessitate additional computational resources and careful consideration of prior distributions, which could introduce subjectivity. Given the complexity of the model parameters, we agree that exploring Bayesian methods could provide valuable insights. We plan to conduct a comparative study in future work to evaluate the performance of various estimation methods, including Bayesian approaches, and assess their respective advantages and disadvantages within our specific context.

We appreciate the reviewer’s suggestion and will consider it a key direction for future research. We believe that such a comparative analysis could strengthen the robustness and generalizability of our findings.

Comment 2: The model mentioned in the paper is specific to the linear Wiener process. It is recommended that the authors consider extending the model to the nonlinear case to handle more complex degradation patterns.

Response 2: We appreciate the reviewer’s suggestion and believe that extending the model to the nonlinear case could significantly broaden the scope of our work. We will incorporate this direction into our future research agenda.

Comment 3: When constructing confidence intervals, the authors adopted the Bootstrap method. It is suggested that the authors discuss other methods for constructing confidence intervals (such as asymptotic methods) and compare the accuracy and efficiency of different methods.

Response 3: Thank you for your excellent comment. We believe that exploring alternative methods for constructing confidence intervals could significantly enhance the methodological depth of our work. While the Bootstrap method was chosen for its flexibility and robustness in our current study, in future research, we will focus on studying the generalized confidence intervals and Bayesian credible intervals for the model parameters and incorporate a comparison of different methods to provide a more comprehensive evaluation of their performance.

Response to Reviewer 2

Comment 1: The paper is well written. However, the following concerns should be addressed. The authors should compare the proposed models with other accelerated degradation models with multiple stress factors.

Response 1: Thanks so much for your constructive comment. In the simulation study and numerical example, the proposed model was compared with the linear random effects multi-stress Wiener accelerated degradation model that does not account for measurement errors. The results indicates that the proposed model exhibits good robustness.

Response to Reviewer 3

Comment 1: The main contribution of this paper is not prominently marked. Please summarize the innovations in the Introduction.

Response 1: We sincerely thank the reviewer for their valuable feedback and for pointing out the need to more prominently highlight the main contributions of our work. We appreciate the opportunity to clarify and emphasize the innovations of our study.

In the revised manuscript, the main contributions of this paper are introduced at the end of the fourth paragraph of the introduction.

Comment 2: How are the parameters of the Wiener process determined and verified?

Response 2: Thank you for your careful comment. The maximum likelihood estimates of the model parameters are derived using the profile likelihood approach. We believe that the methods used for determining and verifying the Wiener process parameters are robust and well-justified.

Comment 3: There are many models of random processes. Why is the Wiener process suitable for this case?

Response 3: Thank you for the valuable comments. The Wiener process is suitable for this study primarily because of its significant advantages in describing random processes and uncertainties in continuous time. Specifically, the increments of the Wiener process are independent and normally distributed, which allows it to naturally capture random fluctuations or noise. This is particularly well-supported in the context of the phenomenon we are studying, where the assumption of Gaussian increments holds. Besides, the Wiener process is continuous and unbiased, with variance increasing linearly over time, which aligns with the core characteristics of the random phenomena we aim to model. Although other accelerated degradation models based on random processes, such as the Poisson process, may be suitable for modeling discrete events, they are not appropriate for capturing the continuous dynamics in our study. Therefore, the Wiener process with random effects is a reasonable and effective choice, accurately reflecting the randomness and uncertainty of the phenomenon.

Comment 4: Recent reliability assessment related work could be referred to, Artificial intelligence enhanced fault prediction with industrial incomplete information, Data-model-linked remaining useful life prediction method with small sample data: A case of subsea valve, Remaining useful life estimation of structure systems under the influence of multiple causes: Subsea pipelines as a case study, Remaining useful life re-prediction methodology based on Wiener process: Subsea Christmas tree system as a case study.

Response 4: We sincerely thank the reviewer for highlighting recent and highly relevant studies in the field of reliability assessment. The studies mentioned by the reviewer provide valuable insights into advanced reliability assessment methods, particularly in industrial applications. In our original manuscript, we focused on foundational and widely recognized literature to establish the theoretical framework for our approach. However, we acknowledge that incorporating these recent works would enhance the context and relevance of our study.

Comment 5: I think there is still a lot of room for improvement in the conclusion part, especially the summary of the work done needs to be further strengthened.

Response 5: We sincerely thank the reviewer for pointing out the need to improve the conclusion section of our manuscript. We agree that the conclusion is a critical part of the paper, and we have carefully revised it to provide a more comprehensive and impactful summary of our work.

Response to Reviewer 4

Comment 1: the degradation model is too usual without much new contribution, a more realistic and complex models should be considered and the engineering background is needed. The engineering mechanism for the failure process is very important for reliability research. Thus, a novel degradation is needed first and the parameter estimation process could be much brief. For example, considering the shock process, multiple degradation indexes, etc.

Response 1: We sincerely appreciate the reviewer’s valuable comments and suggestions. We acknowledge that the degradation model presented in the original manuscript is relatively conventional and may not fully capture the complexity of real-world engineering scenarios. We agree that a more realistic and complex model is needed to better reflect the engineering context and failure mechanisms. We plan to consider complex models as our future work. This is the basis.

Comment 2: What is the application value of the research and the results. A more management view is needed for the conclusion.

Response 2: We sincerely appreciate the reviewer’s valuable feedback regarding the application value of our research. Our research has significant application value in the field of reliability theory and engineering. The proposed model and results can be directly applied to reliability analysis based on Wiener accelerated degradation model, enabling engineers and managers to make more informed decisions. To further illustrate the application value, we have included a case study based on a numerical example. The results demonstrate how the proposed approach can be implemented in real-world scenarios, such as reliability assessment for highly reliable products.

Response to Reviewer 5

Comment 1: More discussions should be provided to prove creativity of the present work in the Introduction part.

Response 1: Thank you for the valuable suggestions regarding the introduction. We fully understand the reviewer’s expectation to highlight the creativity of our work. To better emphasize the innovation of our study, we will add more comparisons between the current research and existing literature in the introduction, and provide a detailed explanation of the unique aspects of our proposed method or theoretical framework in addressing the specific problem.

Comment 2: A nomenclature is suggested since there are too many abbreviations used.

Response 2: Thank you for the valuable suggestion regarding the use of abbreviations. We understand that the frequent use of abbreviations may affect the readability and clarity of the paper. To address this issue, we will follow the reviewer’s suggestion and include a nomenclature section in the paper, listing all abbreviations along with their corresponding definitions. We believe this will help improve the readability of the paper and ensure that readers can more easily understand our research.

Comment 3: Reasonability and generality of the basic assumptions should be clearly discussed, especial A2 and A3.

Response 3: Thank you for the valuable suggestion regarding the discussion of the reasonability and generality of the assumptions. We fully understand the reviewer's concerns about assumptions A2 and A3, particularly regarding their impact on the research results.

A2 is an assumption regarding the consistency of the degradation mechanism. The scale parameter 𝜎 of the Wiener process reflects the 'amplitude' or volatility of the stochastic process. Therefore, to ensure the consistency of the degradation mechanism, a reasonable assumption is that the scale parameters under different test groups are statistically equal.

In practical cases, there exists substantial unit-to-unit variability among degradation processes of different individuals, mainly because of the differences in materials and production conditions. Some studies have shown that such heterogeneities can be well captured by incorporating random effects into the degradation process. Therefore, A3 assume that the drift parameter β to be a random variable, accounting for the unit-to-unit variability.

Comment 4: Transformed variables in A3 was referred to as accelerating variables in the beginning of Section 2.2, please have a check and uniform them.

Response 4: We greatly appreciate the reviewer careful comment. In the revised manuscript, , denotes the i-th stress loading level of the q-th stress factor (i.e., accelerating variable).

Comment 5: “,” is suggested for notations with subscripts, for example, “t_ijk” shoud be “t_i,j,k” since i, j, k might take values more than 10.

Response 5: Thank you for your valuable suggestion regarding the notation format. We understand that using commas in subscripts, such as “ ” can help avoid ambiguity when indices exceed single digits. However, in our manuscript, the indices i, j, and k are consistently defined within a range that does not exceed single-digit values. Therefore, the current notation “ ” does not lead to any confusion in our context.

Comment 6: There are many formulae and numbers in this manuscript, please check them carefully to make sure they are all correct without any typos and mistakes.

Response 6: Thank you for your careful review and valuable suggestion. We fully agree with the importance of ensuring the accuracy of all formulae and numerical values in the manuscript. In response to your comment, we have thoroughly reviewed all the equations, calculations, and numerical data presented in the manuscript to ensure there are no typos or mistakes. We have also double-checked the derivations and cross-verified the results to confirm their correctness. If there are any specific parts of the manuscript that you would like us to revisit or clarify further, please kindly point them out, and we will address them promptly.

Comment 7: In the reference list, please list all authors instead of using “et al.”, and the using of spaces should also be carefully uniformed.

Response 7: Thank you for your careful review and valuable suggestions. We appreciate your attention to detail in improving the quality of our manuscript. In response to your comment, we have revised the reference list to include all authors instead of using “et al.” In addition, we have carefully checked and unified the use of spaces throughout the reference list to ensure consistency.

---

## [Decision Letter · Decision Letter 1]

24 Mar 2025

PONE-D-24-52301R1Reliability assessment model for multiple stress factors accelerated degradation test using a Wiener process with random effectsPLOS ONE

Dear Dr. TANG,

Thank you for submitting your manuscript to PLOS ONE. After careful consideration, we feel that it has merit but does not fully meet PLOS ONE’s publication criteria as it currently stands. Therefore, we invite you to submit a revised version of the manuscript that addresses the points raised during the review process.

**ACADEMIC EDITOR: **One reviewer still has concern regarding the main contribution of this article. It is suggested that the authors strengthen the introduction section.

We look forward to receiving your revised manuscript.

Kind regards,

Qingan Qiu

Academic Editor

PLOS ONE

Journal Requirements:

Additional Editor Comments :

One reviewer still has concern regarding the main contribution of this article. It is suggested that the authors strengthen the introduction section.

Reviewers' comments:

Reviewer's Responses to Questions

**Comments to the Author**

1. If the authors have adequately addressed your comments raised in a previous round of review and you feel that this manuscript is now acceptable for publication, you may indicate that here to bypass the “Comments to the Author” section, enter your conflict of interest statement in the “Confidential to Editor” section, and submit your "Accept" recommendation.

Reviewer #1: All comments have been addressed

Reviewer #2: All comments have been addressed

Reviewer #3: All comments have been addressed

Reviewer #5: All comments have been addressed

2. Is the manuscript technically sound, and do the data support the conclusions?

Reviewer #1: Yes

Reviewer #2: Yes

Reviewer #3: Partly

Reviewer #5: Yes

3. Has the statistical analysis been performed appropriately and rigorously? 

Reviewer #1: Yes

Reviewer #2: Yes

Reviewer #3: Yes

Reviewer #5: Yes

4. Have the authors made all data underlying the findings in their manuscript fully available?

Reviewer #1: Yes

Reviewer #2: Yes

Reviewer #3: Yes

Reviewer #5: Yes

5. Is the manuscript presented in an intelligible fashion and written in standard English?

Reviewer #1: Yes

Reviewer #2: Yes

Reviewer #3: Yes

Reviewer #5: Yes

6. Review Comments to the Author

Reviewer #1: The authors have adequately addressed my comments. I also checked the authors' response to other reviewers' comments. The revision looks well prepared.

Reviewer #2: (No Response)

Reviewer #3: The author has addressed most of the concerns, but the main contribution of this article is still unclear. It is suggested that the author strengthen the introduction section. This article can be accepted after minor revisions.

Reviewer #5: The manuscript is now fine with me since previous comments of the revieweres have been properly addressed.

7. PLOS authors have the option to publish the peer review history of their article (what does this mean?). If published, this will include your full peer review and any attached files.

Reviewer #1: **Yes: **rui peng

Reviewer #2: No

Reviewer #3: No

Reviewer #5: No

---

## [Author Response · Author response to Decision Letter 2]

2 May 2025

Dear Editor and Reviewers:

Thank you for your letter and for the careful comments proposed by the reviewer regarding our revised manuscript (PONE-D-24-52301R1). We have further revised the comments still raised by the reviewers. We have carefully considered all of the feedback, our responses to the reviewers' comments are outlined below:

Response to Reviewers

Reviewer #1: The authors have adequately addressed my comments. I also checked the authors' response to other reviewers' comments. The revision looks well prepared.

Reviewer #2: (No comments)

Reviewer #3: The author has addressed most of the concerns, but the main contribution of this article is still unclear. It is suggested that the author strengthen the introduction section. This article can be accepted after minor revisions.

Response : We sincerely thank the reviewer for the valuable suggestion to emphasize the main contribution of the paper. We have accepted suggestion and summarized the main contributions of the paper based on the opinions of the reviewers; Added specific details to the second to last paragraph of the introduction section on page 3. Correspondingly, we have also made corresponding modifications in the conclusion section. And highlight with blue .

Reviewer #5: The manuscript is now fine with me since previous comments of the revieweres have been properly addressed.

---

## [Decision Letter · Decision Letter 2]

7 May 2025

Reliability assessment model for multiple stress factors accelerated degradation test using a Wiener process with random effects

PONE-D-24-52301R2

Dear Dr. TANG,

We’re pleased to inform you that your manuscript has been judged scientifically suitable for publication and will be formally accepted for publication once it meets all outstanding technical requirements.

Kind regards,

Qingan Qiu

Academic Editor

PLOS ONE

Additional Editor Comments (optional):

The authors have conducted a thorough revision. It can be accepted now.

Reviewers' comments:

Reviewer's Responses to Questions

**Comments to the Author**

1. If the authors have adequately addressed your comments raised in a previous round of review and you feel that this manuscript is now acceptable for publication, you may indicate that here to bypass the “Comments to the Author” section, enter your conflict of interest statement in the “Confidential to Editor” section, and submit your "Accept" recommendation.

Reviewer #3: All comments have been addressed

2. Is the manuscript technically sound, and do the data support the conclusions?

Reviewer #3: Yes

3. Has the statistical analysis been performed appropriately and rigorously? 

Reviewer #3: Yes

4. Have the authors made all data underlying the findings in their manuscript fully available?

Reviewer #3: Yes

5. Is the manuscript presented in an intelligible fashion and written in standard English?

Reviewer #3: Yes

6. Review Comments to the Author

Reviewer #3: I am satisfied with the current version of the manuscript. All comments have been well addressed, the manuscript is now acceptable.

7. PLOS authors have the option to publish the peer review history of their article (what does this mean?). If published, this will include your full peer review and any attached files.

Reviewer #3: No

---

## [Editor Report · Acceptance letter]

PONE-D-24-52301R2

PLOS ONE

Dear Dr. TANG,

I'm pleased to inform you that your manuscript has been deemed suitable for publication in PLOS ONE. Congratulations! Your manuscript is now being handed over to our production team.

Kind regards,

on behalf of

Dr. Qingan Qiu

Academic Editor

PLOS ONE